# A wheat resistosome defines common principles of immune receptor channels

Alexander Förderer[1,2,7], Ertong Li[1,2,7], Aaron W. Lawson[2,7], Ya-nan Deng[3,4,7], Yue Sun[5], Elke Logemann[2], Xiaoxiao Zhang[5], Jie Wen[5], Zhifu Han[5], Junbiao Chang[6], Yuhang Chen[3,4✉], Paul Schulze-Lefert[2✉] & Jijie Chai[1,2,5✉]

Plant intracellular nucleotide-binding leucine-rich repeat receptors (NLRs) detect pathogen effectors to trigger immune responses[1]. Indirect recognition of a pathogen effector by the dicotyledonous *Arabidopsis thaliana* coiled-coil domain containing NLR (CNL) ZAR1 induces the formation of a large hetero-oligomeric protein complex, termed the ZAR1 resistosome, which functions as a calcium channel required for ZAR1-mediated immunity[2–4]. Whether the resistosome and channel activities are conserved among plant CNLs remains unknown. Here we report the cryo-electron microscopy structure of the wheat CNL Sr35[5] in complex with the effector AvrSr35[6] of the wheat stem rust pathogen. Direct effector binding to the leucine-rich repeats of Sr35 results in the formation of a pentameric Sr35–AvrSr35 complex, which we term the Sr35 resistosome. Wheat Sr35 and *Arabidopsis* ZAR1 resistosomes bear striking structural similarities, including an arginine cluster in the leucine-rich repeats domain not previously recognized as conserved, which co-occurs and forms intramolecular interactions with the 'EDVID' motif in the coiled-coil domain. Electrophysiological measurements show that the Sr35 resistosome exhibits non-selective cation channel activity. These structural insights allowed us to generate new variants of closely related wheat and barley orphan NLRs that recognize AvrSr35. Our data support the evolutionary conservation of CNL resistosomes in plants and demonstrate proof of principle for structure-based engineering of NLRs for crop improvement.

Plant nucleotide-binding leucine-rich repeat receptors (NLRs) are intracellular receptors that play a key role in the plant innate immune system by sensing the presence of pathogen effectors delivered inside plant cells during pathogenesis through direct or indirect recognition[1,7]. Activation of plant NLRs generally leads to an array of immune responses, often linked to rapid host cell death at sites of attempted pathogen infection. Structural and functional homologues of plant NLRs evolved from independent events for intracellular non-self-perception in animal innate immunity and are characterized by their conserved central nucleotide-binding and oligomerization domains (NODs)[8]. Plant NLRs can be broadly separated into two classes: CNL with an N-terminal coiled-coil domain and TNL with an N-terminal Toll/interleukin 1 receptor (TIR) domain. Among the flowering plants dicots typically possess both receptor classes, whereas monocots, including cereals, encode only CNL receptors[9].

Wheat stem rust caused by fungal infection with *Puccinia graminis* f. sp. *tritici* (*Pgt*) threatens global wheat production[10], and the emergence of widely virulent *Pgt* strains, including the Ug99 lineage, has motivated the search for stem rust resistance in wheat germplasm and wild relatives over the past two decades. This resulted in the isolation of 11 phylogenetically related stem rust resistance (*Sr*) genes that belong to a clade of grass CNLs, all of which confer strain-specific immunity[5,11–18] ('clade I' CNLs defined in ref. [18]). The mildew resistance locus A (MLA) receptors of the wheat sister species barley also belong to this group of grass CNLs and share strain-specific immunity with *Sr* genes[18]. *Sr35* was first identified in a landrace of the *Triticum urartu* relative *Triticum monococcum* (einkorn) and confers immunity to *Pgt* Ug99 in bread wheat when transferred as a transgene[5]. However, because *Sr35* was absent in the diploid A genome of the wild ancestor of wheat, *T. urartu*, it was initially absent in hexaploid bread wheat (*Triticum aestivum*). *Sr35* resistance has been linked to the recognition of the *Pgt* effector *AvrSr35*[6], but until now, it has remained inconclusive whether Sr35 receptor-mediated host cell death is driven by direct physical interaction with the AvrSr35 effector[6,19].

## Cryo-electron microscopy of the Sr35 resistosome

To purify Sr35, we expressed the protein alone or together with AvrSr35 in Sf21 insect cells. Unexpectedly, cell death was observed when the receptor was co-expressed with *AvrSr35* (Extended Data Fig. 1a),

[1]Institute of Biochemistry, University of Cologne, Cologne, Germany. [2]Max Planck Institute for Plant Breeding Research, Cologne, Germany. [3]State Key Laboratory of Molecular Developmental Biology, Institute of Genetics and Developmental Biology, Chinese Academy of Sciences, Beijing, China; Innovative Academy of Seed Design, Chinese Academy of Sciences, Beijing, China. [4]College of Advanced Agricultural Sciences, University of Chinese Academy of Sciences, Beijing, China. [5]Beijing Advanced Innovation Center for Structural Biology, Tsinghua-Peking Joint Center for Life Sciences, Center for Plant Biology, School of Life Sciences, Tsinghua University, Beijing, China. [6]Henan Key Laboratory of Organic Functional Molecules and Drug Innovation, Henan Normal University, School of Pharmaceutical Sciences, Zhengzhou University, Zhengzhou, China. [7]These authors contributed equally: Alexander Förderer, Ertong Li, Aaron W. Lawson, Ya-nan Deng. ✉e-mail: yuhang.chen@genetics.ac.cn; schlef@mpipz.mpg.de; chai@mpipz.mpg.de

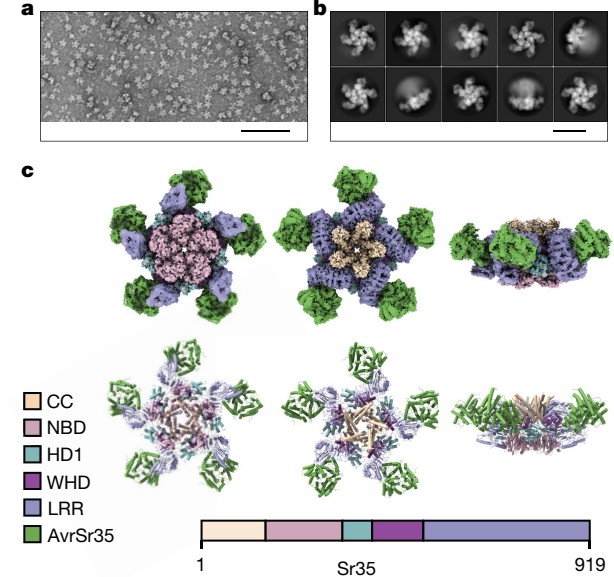

**Fig. 1 | 3D reconstruction of the Sr35 resistosome. a**, Negative staining of purified Sr35 in complex with AvrSr35 (Sr35 resistosome). Star-shaped particles were enriched by affinity purification and size-exclusion chromatography. Monodisperse Sr35 resistosomes have an average size of approximately 24 nm. Scale bar, 100 nm. **b**, 2D classifications of the Sr35 resistosome particles from the cryo-EM sample. Particles show preferential orientations for bottom and top views. Fewer, but sufficient, particles are in side view. Scale bar, 20 nm. **c**, Cryo-EM density map with 3 Å resolution (top) and the finally refined structure model (bottom) of the Sr35 resistosome shown in three different orientations. AvrSr35 is coloured green and Sr35 domains are shown according to the colour codes in the inset panel.

suggesting that Sr35 and its effector are sufficient to mediate this immunity-associated response in insect cells in the absence of other plant proteins. To circumvent cell death activation for protein purification, we introduced substitutions in the N-terminal residues L15E/L19E (Sr35[L15E/L19E]), which are predicted to be essential for Sr35 membrane association by analogy with the ZAR1 resistosome[3]. Mutational analysis of the corresponding N-terminal residues of the tomato CNL NRC4 has been shown to abrogate cell death activity in *Nicotiana benthamiana*[20]. Indeed, the Sr35[L15E/L19E] substitutions markedly reduced *Sr35*-induced cell death in insect cells (Extended Data Fig. 1a and Supplementary Table 1). Using affinity-tagged Sr35[L15E/L19E] co-expressed with affinity-tagged AvrSr35, we were able to enrich the Sr35–AvrSr35 complex for subsequent separation of potential receptor complex isoforms and correctly folded receptor complexes by size-exclusion chromatography (Extended Data Fig. 1b,c). The affinity-purified protein complex was eluted with a broad peak with a maximum UV absorbance at 65 ml elution volume exceeding 669 kDa (66 ml) of our largest protein marker. Individual fractions were analysed via negative staining and a large number of star-shaped particles with fivefold symmetry were identified (fractions at 60–69 ml; Fig. 1a). The most monodisperse fractions were pooled and used for cryo-electron microscopy (cryo-EM) analysis.

We analysed the Sr35–AvrSr35 complex sample by cryo-EM (Extended Data Fig. 1) using a total of 1,608,441 individual particles for reference-free two-dimensional (2D) classification (Fig. 1b). After three-dimensional (3D) classification, a subset of 230,485 particles was used for reconstruction, yielding a density map of 3.0 Å (Fig. 1c, top). Despite the high resolution of up to 2.5 Å in the centre of the complex, the local resolution decreased towards the outer edge to approximately 4 Å (Extended Data Fig. 1f), indicating that the outer region of the complex is more flexible. To compensate for this decreased resolution, a local mask was used for the outer region, yielding a local density map with a resolution of 3.33 Å (Extended Data Fig. 1f). Both density maps were used for model building (Fig. 1c, bottom).

The final 3D reconstruction of the Sr35–AvrSr35 complex contains five receptor protomers, each bound to one effector molecule. The reconstruction revealed a star-shaped structure, similar to the ZAR1 resistosome[3], that we termed the Sr35 resistosome. As in the ZAR1 resistosome, five Sr35 NOD modules define the base of the circular protomer arrangement, and a helical barrel formed by the five coiled-coil domains is buried at the centre. Unlike ZAR1, the leucine-rich repeat (LRR) domains at the outer region do not pack against each other in the Sr35 resistosome, which might explain why this region is more flexible. AvrSr35 adopts an exclusively helical fold (Extended Data Fig. 2). A 3D structure homology search using the server DALI[21] showed that there are no other known protein structures sharing the AvrSr35 fold. Five AvrSr35 proteins bind exclusively to the C-terminal part of the LRR domains in the complex.

## Oligomerization of the Sr35 resistosome

The central NOD module of plant NLRs is subdivided into a nucleotide-binding domain (NBD), helical domain 1 (HD1) and winged helical domain (WHD). ATP/dATP has been shown to be important for ZAR1 oligomerization as it stabilizes the active conformation of ZAR1 via its interaction with the WHD in the NOD module. There is an unambiguous cryo-EM density at the predicted nucleotide-binding site between the HD1 and NBD domains that is unfilled by Sr35 and AvrSr35. An ATP molecule fits well into this cryo-EM density. The modelled ATP is nested in a groove formed by HD1 and NBD. The short α-helix that mediates interprotomer interaction (Fig. 2a,c) also caps the ATP molecule. In contrast to that of ZAR1, the ATP in Sr35 does not directly contact the WHD. Instead, the γ-phosphate group of the ATP forms a bidentate hydrogen bond with Sr35 NBD[R157] and NBD[R311] (Fig. 2c). The latter also forms a hydrogen bond with Sr35 WHD[S420] (Fig. 2c), showing an indirect coupling of the ATP γ-phosphate group with the WHD of Sr35.

Similar to the ZAR1 resistosome, NBD–NBD contacts contribute to Sr35 protomer packing (Fig. 2a,d). Sr35 NBD[Y244] from one protomer packs tightly against Sr35 NBD[R259] and Sr35 NBD[Y263] from an adjacent protomer (Fig. 2d). Additionally, a hydrogen bond is established between Sr35 NBD[Y244] and Sr35 NBD[R259]. Of note, the coiled-coil domain of Sr35 contributes considerably to the interprotomer interactions (Fig. 2a): the C-terminal half of α4-helix from one protomer packs against the C-terminal sides of α2- and α4-helices of the neighbouring coiled-coil protomer. At the centre of this interface in the coiled-coil is Tyr141 (CC[Y141]), which makes extensive hydrophobic contacts with Sr35 CC[L42], CC[M43], CC[L47] and CC[W65] (Fig. 2e). Moreover, CC[Y141] participates in a hydrogen bonding triad together with CC[R140] from the same and CC[E39] from the neighbouring protomer (Fig. 2e). As previously reported[22], the long linker region between the coiled-coil and NBD domain is also involved in mediating oligomerization of the Sr35 resistosome.

To functionally test the requirements for these interactions in mediating the assembly of the Sr35 hetero-oligomeric complex, we introduced amino acid substitutions into the receptor and assessed their impact on Sr35-mediated cell death using a luciferase (LUC) activity assay in wheat protoplasts[23] prepared from a genotype that does not recognize AvrSr35 (cultivar 'Chinese Spring'). In this protoplast transfection assay, the relative (to empty vector, EV) luminescence of the LUC reporter is an indicator of cell viability. Cotransfection of *Sr35*, *AvrSr35* and the *LUC* reporter resulted in a near complete loss of luminescence signal, indicating massive cell death of the protoplasts and suggesting receptor activation by AvrSr35 (Fig. 2g). Consistent with the insect cell data described above, wheat protoplasts co-expressing *Sr35[L15E/L19E]* and *AvrSr35* displayed luminescence levels that were comparable to those co-expressing *EV* and *AvrSr35* constructs, indicating

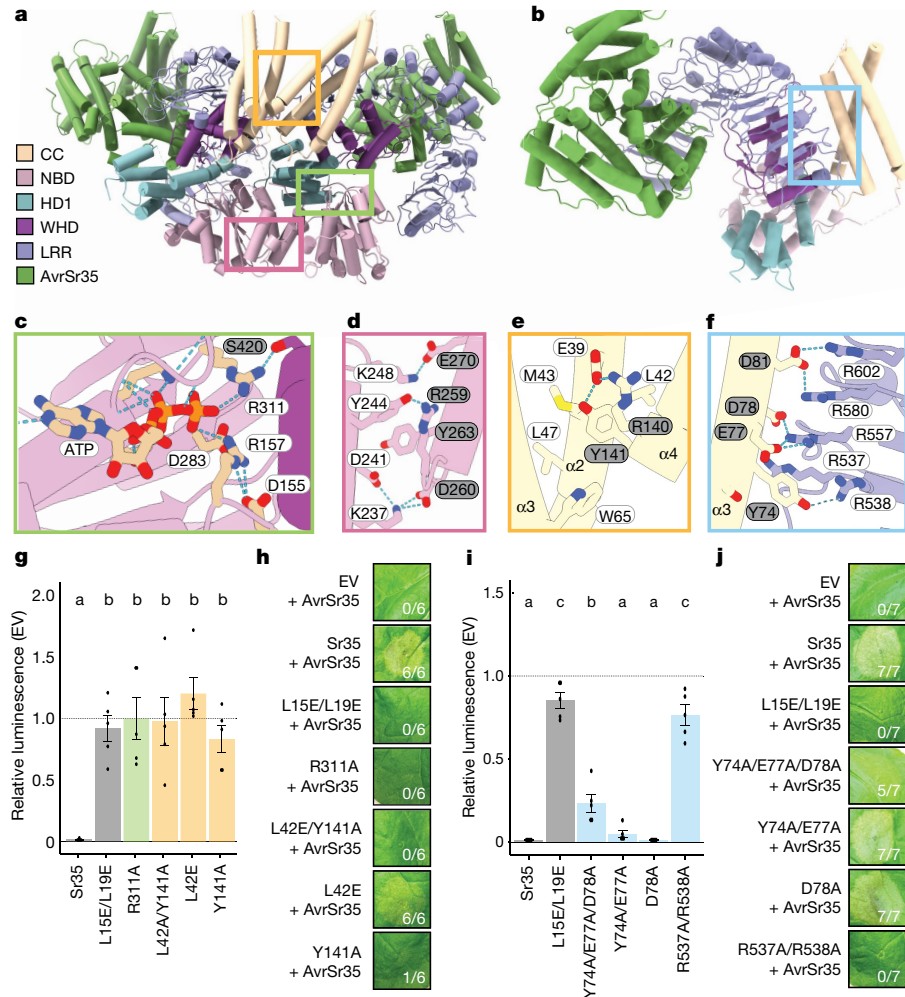

**Fig. 2 | Assembly of the Sr35 resistosome. a**, Sr35 resistosome showing a lateral dimer. Boxes in green, yellow and pink indicate positions of the zoomed views in **c**–**f**. Sr35 domains and AvrSr35 coloured according to the inset panel. **b**, Structure of one Sr35 protomer in complex with AvrSr35. Colour codes as in **a**. The blue box indicates position of structural detail in **f**. **c**, Structural detail of ATP binding in one protomer. Note the specific hydrogen bond of R311 with the γ-phosphate group of ATP at a 2.8 Å distance. Grey and white residue labels correspond to NBD and WHD residues, respectively. **d**, Structural detail of the interface between NBDs of a lateral dimer. Dashed lines represent polar interactions. Grey and white residue labels correspond to two neighbouring protomers from the pentamer. **e**, Structural detail of interface between two coiled-coil (CC) protomers. **f**, Structural detail of coiled-coil and LRR domain intramolecular packing in one Sr35 protomer. Acidic residues in the CC$^{EDIVD}$ form salt-bridges with basic Arg (R) residues of the LRR$^{R-cluster}$. **g**, Cotransfection of *Sr35* and *Sr35* mutants with *AvrSr35* in wheat protoplasts. Relative luminescence as readout for cell death. EV treatment defined the relative baseline (mean ± s.e.m.; $n = 5$). Test statistics derived from analysis of variance (ANOVA) and Tukey post hoc tests ($P < 0.05$). Exact $P$ values for all protoplast plots are provided in Supplementary Table 3. Bar colours as box colours in **c** and **d**. **h**, Tobacco cell death data of *Sr35* and *Sr35* mutants with *AvrSr35*. **i**, Wheat protoplast data of EDIVD and R-cluster mutants. Experiment and statistics as in **g**. Bar colours as box colour in **f**. **j**, *Nicotiana benthamiana* cell death data of EDVID and R-cluster mutants. Representative data in **h** and **j** shown from seven replicates and scored for leaf cell death.

that the cell death activity of the Sr35$^{L15E/L19E}$ receptor is substantially impaired (Fig. 2g). A similarly drastic loss of receptor-mediated cell death activity was observed with substitutions predicted to affect coiled-coil interprotomer interactions (Y141A, L42E and L42E/Y141A) or the ATP-binding site (R311A) (Fig. 2g) (raw data of all protoplast measurements are provided in Supplementary Table 2).

To corroborate the data from wheat protoplasts, we used *Agrobacterium tumefaciens*-mediated transient gene expression of *Sr35* and *AvrSr35* in *N. benthamiana* leaves. Co-expression of *Sr35* and *AvrSr35*, but not *AvrSr35* plus EV, resulted in cell death in the *Agrobacterium*-infiltrated area (Fig. 2h). By contrast, cell death was abolished when *AvrSr35* was co-expressed with the *Sr35* mutants predicted to perturb Sr35 oligomerization (Fig. 2h), with the exception of Sr35$^{L42E}$, which showed residual cell death activity only in *N. benthamiana* (full versions of all tobacco agroinfiltrations are provided in Supplementary Figs. 1–8). In planta, protein levels of wild-type Sr35 and all

receptor mutants tested were comparable, indicating that these substitutions do not render the receptor unstable (Extended Data Fig. 3a, and full versions of all blots are provided in Supplementary Figs. 9–11). Together, these data strongly suggest that the residues mediating Sr35 oligomerization in the cryo-EM structure are necessary for cell death activity in wheat and heterologous *N. benthamiana*.

A conserved sequence in the coiled-coil domain, long known as the 'EDVID (Glu-Asp-Val-Ile-Asp) motif' that is present in approximately 38% of *Arabidopsis* CNLs[24] and first described in the potato CNL Rx, is used to group CNLs with or without this motif[24,25]. In the cryo-EM structure of the Sr35 resistosome, the EDIVD motif (Glu-Asp-Ile-Val-Asp) and the adjacent Sr35 Y[74] mediate the packing of the LRR domain against the coiled-coil domain. Acidic residues from the motif form strong contacts with five arginine residues in the LRR domain (LRR$^{R537}$, LRR$^{R538}$, LRR$^{R557}$, LRR$^{R580}$ and LRR$^{R602}$). These contacts comprise two bidentate salt bonds and a cation–π interaction (Fig. 2b,f). The extensive contacts

in this region are further reinforced by hydrogen bonding and van der Waals contacts. These arginine residues are each separated by one iteration of the LRR motif, resulting in their spatial separation along the Sr35 amino acid sequence (Extended Data Fig. 4a). As previously noted[26], the cryo-EM structure of the ZAR1 resistosome shows that similar intramolecular interactions exist between arginine residues in the ZAR1 LRR and 'EDVID'. In both resistosomes the respective arginine residues cluster together and form a positively charged surface patch (Extended Data Fig. 4b). We therefore term this resistosome region LRR[R-cluster]. Location of the arginine residues in different repeats of the LRR domain explains why the conservation of the LRR[R-cluster] had remained unnoticed. A sequence alignment of CNLs shows that the LRR[R-cluster] is conserved and co-occurs with the EDVID motif (Extended Data Fig. 4a).

To test whether the LRR[R-cluster] is necessary for *Sr35*-mediated cell death, we substituted residues from the interface between the arginine cluster and the EDIVD motif and assessed the impact of these mutations on cell death activity using the wheat protoplast and *N. benthamiana* leaf assays described above. Simultaneous mutations of LRR[R537A/R538A] in the LRR[R-cluster] essentially abolished cell death activity (Fig. 2i,j). Similarly, a triple substitution in the Sr35 EDIVD motif, including the adjacent Sr35 Y[74], (Y74A/E77A/D78A) reduced or abolished Sr35 cell death activity in protoplasts and *N. benthamiana*, without affecting NLR stability (Extended Data Fig. 3b). These observations suggest that the co-occurrence of the EDVID motif and LRR[R-cluster] is an evolutionarily conserved stabilization mechanism of CNL resistosomes. As the EDVID–LRR[R-cluster] interactions are also present in the inactive ZAR1 and AlphaFold2-modelled Sr35 monomers and an extensive fold switching occurs in the coiled-coil domain during receptor activation (Extended Data Fig. 4c), these intramolecular interactions may be transiently disrupted to allow α1-helix to flip.

## Channel activity of the Sr35 resistosome

Albeit having only 28.4% sequence conservation and although the α1-helix region of Sr35, whose equivalent in ZAR1 resistosome forms a funnel-shaped structure, is not well defined, the structures of the wheat Sr35 and *Arabidopsis* ZAR1 resistosomes are highly similar (Extended Data Fig. 5). We thus reasoned that the two complexes might share channel activity. To test this conjecture we used an assay previously established in *Xenopus laevis* (*Xenopus*) oocytes[4] to assess potential channel activity of the Sr35 resistosome. Co-expression of *Sr35* and *AvrSr35*, but not either alone, generated currents as recorded by two-electrode voltage-clamp (Fig. 3a,b), suggesting that assembly of the Sr35 resistosome is required for the currents. In strong support of this conclusion, two *Sr35* mutants that impaired the interaction with AvrSr35 and abolished AvrSr35-dependent cell death activity of the receptor in planta (Sr35[R730D/R755Q] and Sr35[W803L/K754G]; see below), lost their ability to produce currents in oocytes (Fig. 3c). In agreement with the data on cell death *in planta* and insect cells, co-expression of the α1-helix substitution mutant *Sr35[L15E/L19E]* with *AvrSr35* did not mediate currents in oocytes (Fig. 3c). Substitutions affecting the acidic inner lining of the funnel formed by α1-helices in ZAR1[E11A] have been shown to abolish cell death in planta and channel activities in oocytes[3,4]. Unexpectedly, both Sr35 resistosome channel and cell death activities were tolerant to these analogous acidic residue substitutions (Sr35[E17A/E22A]) (Fig. 3c–e and Extended Data Fig. 3c) (raw data of all oocytes measurements are provided in Supplementary Table 4).

*Xenopus* oocytes express endogenous calcium-gated chloride channels (CaCC); thus, the currents detected in this assay could be confounded by the activity of these native channels. However, the addition of the CaCC inhibitor A01 only partially blocked the currents in *Xenopus* oocytes (Fig. 3b) and cotreatment with A01 and the calcium channel blocker LaCl₃ was required for complete inhibition of the electrical activity (Fig. 3b). Together, these results indicate that the Sr35

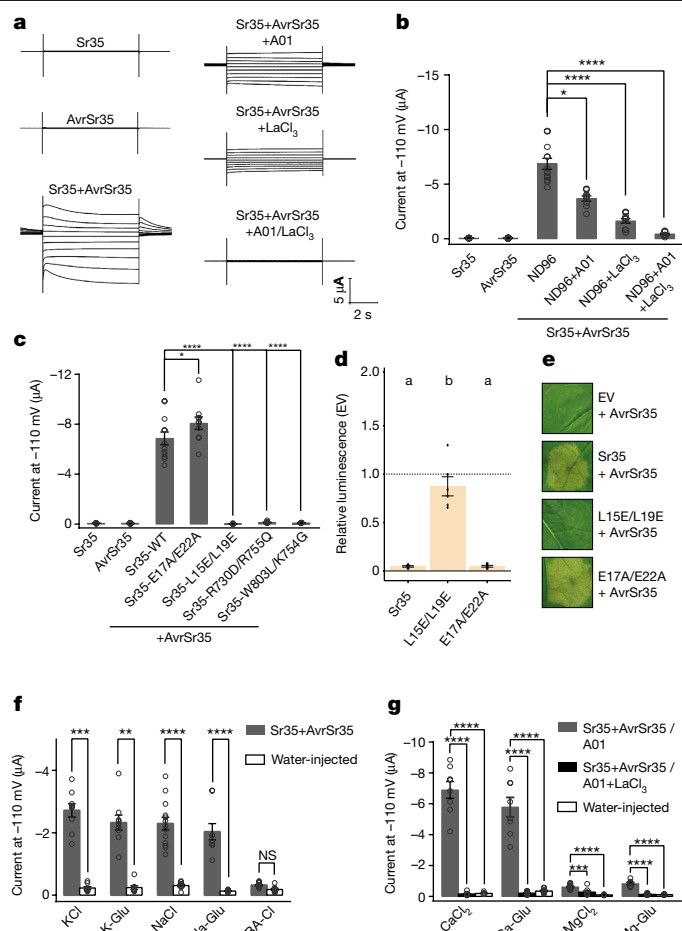

**Fig. 3 | The Sr35 resistosome forms a Ca²⁺-permeable non-selective cation channel. a**, Representative measurements from two-electrode (TEVC) recordings from *Xenopus* oocytes expressing *Sr35*, *AvrSr35* and *Sr35/AvrSr35*. Effects of CaCCinh-A01 (Ca²⁺-activated chloride channel inhibitor) and LaCl₃ (Ca²⁺ channel blocker) on the Sr35-mediated currents in ND96 solution (96 mM NaCl, 2.5 mM KCl, 1 mM MgCl₂, 1.8 mM CaCl₂, 5 mM HEPES, pH 7.6). Current traces shown at different voltages from −110 mV to +70 mV in 20 mV increments and current amplitudes at −110 mV. **b**, Quantitative measurements of data as in **a**. **c**, Structure-based mutagenesis of Sr35 residues at the interface between the LRR domain of Sr35 and AvrSr35, and Sr35 α1-helix. TEVC recordings in ND96 solution, and current amplitudes at −110 mV. **d**, Wheat protoplast data of Sr35 mutations at α1-helix. Relative luminescence as readout for cell death. EV treatment defined the relative baseline (mean ± s.e.m.; *n* = 5). Test statistics derived from ANOVA and Tukey post hoc tests (*P* < 0.05). Exact *P* values are provided in Supplementary Table 3. **e**, Tobacco cell death data of Sr35 and Sr35 channel mutants. Representative data shown from a minimum of three replicates. **f**, The Sr35 channel is selective for cations. TEVC recordings performed in various solutions, including KCl (96 mM), K-gluconate (96 mM), NaCl (96 mM), Na-gluconate (96 mM) and TBA-Cl (96 mM). **g**, Cationic currents of CaCl₂, Ca-Glu, MgCl₂ and Mg-Glu in the presence of CaCCinh-A01 and CaCCinh-A01+LaCl₃. Data are mean ± s.e.m., *n* ≥ 8 (**b**,**c**,**f**,**g**). *P ≤ 0.05, **P ≤ 0.01, ***P ≤ 0.001, ****P ≤ 0.0001, one-way ANOVA analyses and Tukey post hoc test in **b**, **c** and **g**, and two-sided Student's *t*-tests in **f**.

resistosome may contribute to mixed currents in *Xenopus* oocytes, possibly via Sr35 resistosome calcium channel activity.

To test whether the Sr35 resistosome can function as a non-selective cation channel, we tested cation flux in the presence of monovalent solutions of potassium and sodium chloride salts (KCl, NaCl). Similar to the ZAR1 resistosome, co-expression of *Sr35* and *AvrSr35* increased cation flux in oocytes, which was retained for potassium and sodium salts with the immobile gluconate counter-ion (K-Glu, Na-Glu). By contrast,

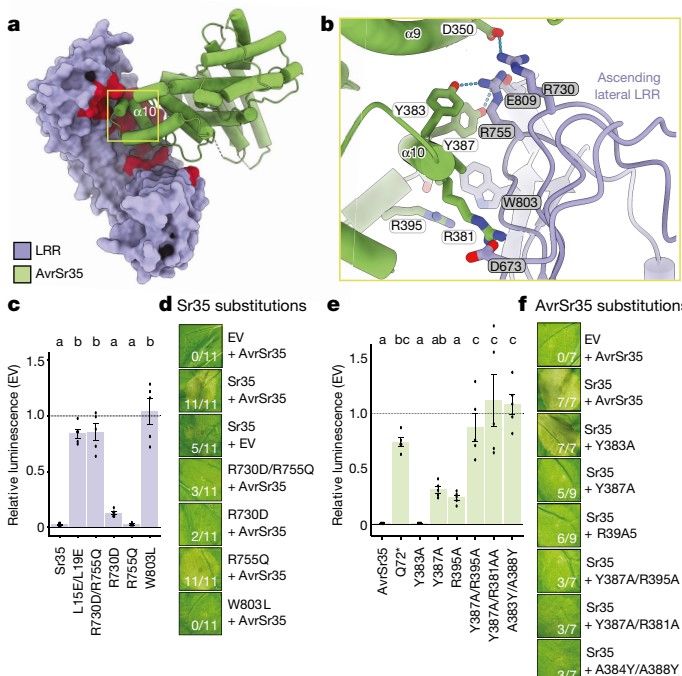

**Fig. 4 | Direct AvrSr35 effector recognition is mediated by the Sr35 LRR domain. a**, Interface between Sr35 LRR and AvrSr35. Red colour indicates the critical LRR residues within 5 Å from AvrSr35. **b**, Structural detail of Sr35 receptor and AvrSr35 effector interface. Dashed lines represent polar interactions. Grey and white residue label boxes correspond to Sr35 and AvrSr35 sidechains, respectively. **c**, Cotransfection of *Sr35* LRR mutants with *AvrS35* in wheat protoplasts. Relative luminescence as readout for cell death. EV treatment defined the relative baseline (mean ± s.e.m.; *n* = 5). Test statistics derived from ANOVA and Tukey post hoc tests (*P* < 0.05). Exact *P* values for all protoplast plots are provided in Supplementary Table 3. Bar colours correspond to box colours in **b**. **d**, *Nicotiana benthamiana* cell death data of *Sr35* LRR mutations at the receptor–effector interface. Representative data are shown from 11 replicates and scored for leaf cell death. **e**, Cotransfection of *Sr35* with *AvrS35* mutants in wheat protoplasts. Experimental layout and statistics as in **c** (mean ± s.e.m.; *n* = 5). Bar colours as domain colours in **a**. **f**, *Nicotiana benthamiana* cell death data of *AvrSr35* mutants co-expressed with *Sr35*. Representative data are shown from nine replicates and scored for leaf cell death.

we observed only residual ion flux when a chloride salt of the immobile tetrabutylammonium was used (TBA-Cl) (Fig. 3f). A comparison of the divalent ions $Ca^{2+}$ and $Mg^{2+}$ ($MgCl_2$, $CaCl_2$) combined with the *Sr35* and *AvrSr35* co-expression in oocytes, showed that ion flux was significant for calcium but not magnesium (Fig. 3g). This finding supports the conclusion that the Sr35 resistosome is permeable to calcium. Although our collective data strongly suggest that the Sr35 resistosome functions similarly to the ZAR1 resistosome by forming a non-selective calcium channel, the channel activity of the Sr35 resistosome is tolerant to substitutions of acidic residues predicted to line the inner surface of the channel. Thus, we cannot exclude the possibility that the very N terminus of the Sr35 resistosome (residues 1–21) is structurally and functionally distinct from that of the ZAR1 resistosome.

## Direct recognition of AvrSr35 by Sr35

In the cryo-EM structure, AvrSr35 binds to the very C-terminal part of the Sr35 LRR domain, supporting a direct recognition mechanism of AvrSr35 by Sr35 (Fig. 4a). AvrSr35 is much larger than many other pathogen effectors, but only a small portion of the protein is in contact with the Sr35 LRR through charge and shape complementarity (Fig. 4a and Extended Data Fig. 6a,b). Nearly all residues that contribute to recognizing AvrSr35 are

from the ascending lateral side of the last eight LRRs, and many of the residues interact with a single helix (α10) of AvrSr35. AvrSr35[Y383], AvrSr35[A384], AvrSr35[Y387] and AvrSr35[A388] from one α10 side are located at the centre of the Sr35–AvrSr35 interface and make extensive contacts with their respective neighbouring residues in Sr35 (Fig. 4b). Several residues in the loop region located C-terminal to α10 form hydrophobic contacts with Sr35[W919]. Similar interactions are also made between AvrSr35[R381] in the N-terminal side of α10 and Sr35. In addition to the hydrophobic and van der Waals interactions, a large network of hydrogen bonds also mediates the Sr35–AvrSr35 interface, supporting specific recognition of AvrSr35 by Sr35 (individual contacts provided in Fig. 4b).

To functionally verify the Sr35–AvrSr35 interaction, we first substituted R730, R755 and W803 in Sr35 with their counterparts in the Sr35 homologue of wheat cultivar Chinese Spring[27] (here denoted *TaSh1*), which shares 86.5% sequence identity with Sr35 but is derived from a wheat cultivar susceptible to *Pgt* strains encoding *AvrSr35*[14]. These W803L or R730D substitutions strongly and weakly suppressed *Sr35*-mediated cell death activity, respectively, when co-expressed with *AvrSr35* in wheat protoplasts (Fig. 4c). By contrast, R755Q had no detectable effect on *Sr35*-induced cell death, but its combination with R730D resulted in a complete loss of cell death in wheat protoplasts (Fig. 4c). Similar results were obtained when these *Sr35* mutants were assayed in *N. benthamiana* (Fig. 4d and Extended Data Fig. 3d). These data support the Sr35–AvrSr35 interaction in the cryo-EM structure and explain why *TaSh*1 in susceptible cultivar Chinese Spring is unable to recognize AvrSr35. To further verify specific AvrSr35 recognition by Sr35, we made the following substitutions in the fungal effector at their interface: Y383A, Y387A, R395A, Y387A/R395A, Y387A/R381A, A384Y/A388Y, all of which either form hydrogen bonds or salt-bridges with the Sr35 LRR (Fig. 4b). Similar to the Q72* premature stop codon mutant of AvrSr35 (Fig. 4e)[6], the mutations Y387A/R395A, Y387A/R381A and A384Y/A388Y abolished *Sr35*-induced cell death in wheat protoplasts and *N. benthamiana* (Fig. 4e,f and Extended Data Fig. 3e). By contrast, single mutations of Y387A and R395A only partially abolished effector-triggered receptor activation (Fig. 4e,f and Extended Data Fig. 3e), and several other single mutations of AvrSr35 (Extended Data Fig. 6b) had no effect, suggesting that much of the AvrSr35–Sr35 interface is resilient to disruption by single amino acid substitutions.

## Sr35 receptor activation by steric clash

We made structural predictions of inactive, monomeric Sr35 using Alpha-Fold2 (ref. [28]). In these predictions, structures of all individual domains were highly similar to those in the Sr35 resistosome and the LRR domain in particular was accurately predicted (Extended Data Fig. 7a). Although some predictions were a close match with the domain organization of Sr35 in the resistosome, other individual predictions showed striking differences in the domain organization of NOD module (NBD–HD1 relative to WHD) (Extended Data Fig. 7b). These predictions shared the relative domain organization of inactive, monomeric ZAR1 and other inactive NLR structures[29], and most likely represent an inactive Sr35 structure. Modelling of AvrSr35 onto the LRR domain of the predicted structure of inactive Sr35 shows substantial overlap between the effector and Sr35 NBD (Extended Data Fig. 8). This is reminiscent of ZAR1 activation, which occurs through an allosteric mechanism involving a 'steric clash' with the NBD[2,3]. Comparison of Sr35 and ZAR1 resistosomes suggests that this 'steric clash' mechanism is likely to be conserved in CNLs. AvrSr35 binding dislodges the NBD, allowing subsequent nucleotide exchange for further ATP-triggered allosteric changes in the receptor and assembly of the Sr35 resistosome. Together, these results support a conserved allosteric mechanism underlying activation of the Sr35 and ZAR1 resistosomes. Ligand binding to the ascending lateral side of the LRR domain was also seen in the structures of the TNL RPP1 (ref. [30]) and Roq1 (ref. [31]) resistosomes (Extended Data Fig. 9), suggesting that the ligand binding mechanism may be conserved in plant NLRs[29].

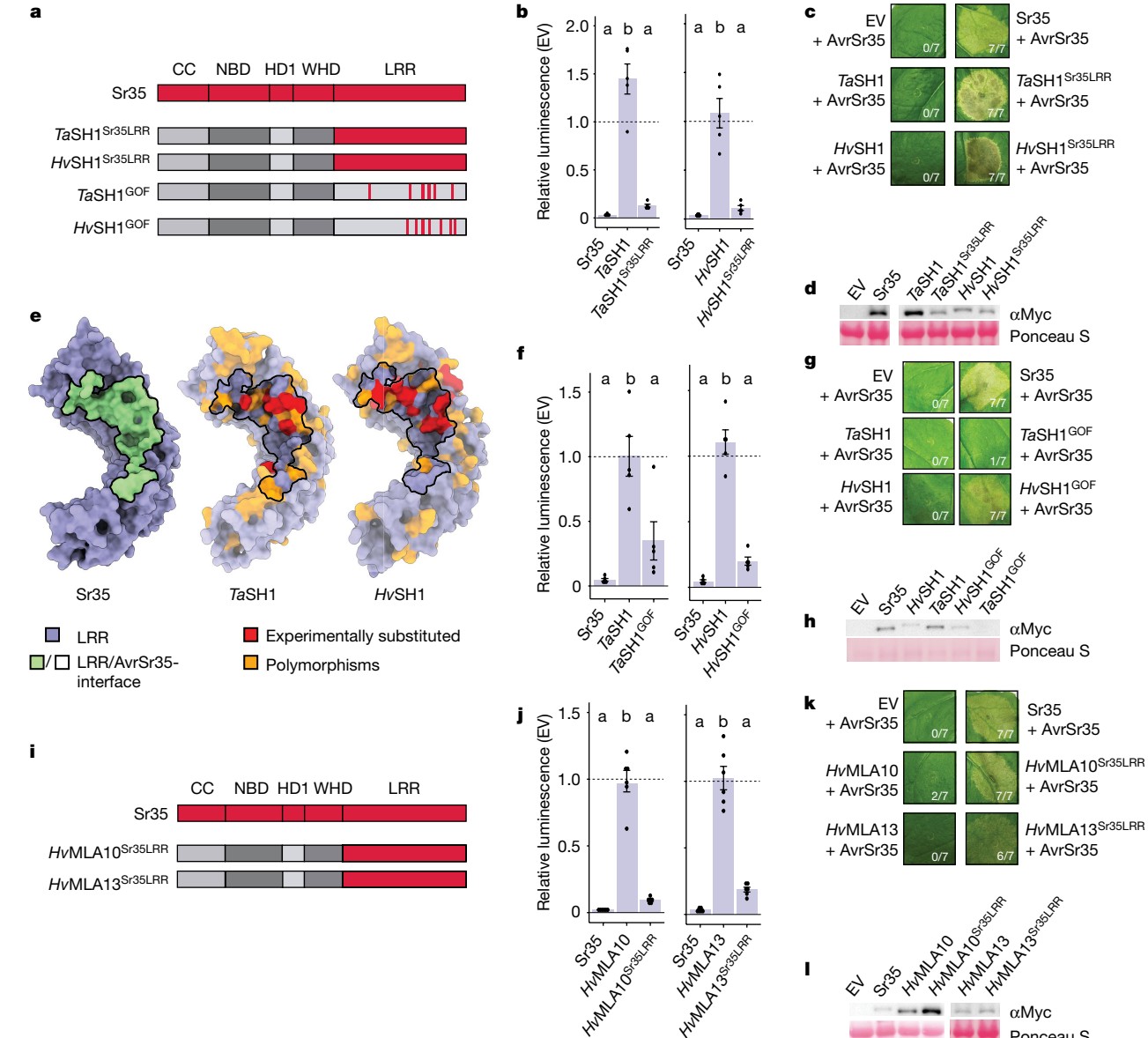

**Fig. 5 | Structure-guided neofunctionalization of orphan CNLs and MLA receptor hybrids. a**, Illustration of Sr35 domain structure and hybrid receptors made from Sr35 homologues (*Sh*) in bread wheat (*Triticum aestivum*; *TaSh1*) and barley (*Hordeum vulgare*; *HvSh1*). Sr35[LRR] (red) substitutes *TaSH1*[LRR] and *HvSH1*[LRR] (*TaSH1*[Sr35LRR] and *HvSH1*[Sr35LRR]). GOF receptor variants (*TaSh1*[GOF] and *HvSh1*[GOF]) were derived from sequence polymorphisms between Sr35, *TaSH1* and *HvSH1*. **b**, Wheat protoplast transfections of *TaSh1*[Sr35LRR], *HvSh1*[Sr35LRR] and controls co-expressed with *AvrSr35*. EV treatment defined the relative baseline (mean ± s.e.m.; *n* = 6). Test statistics derived from ANOVA and Tukey post hoc tests (*P* < 0.05). Exact *P* values for all protoplast plots are provided in Supplementary Table 3. **c**, Tobacco cell death of *TaSh1*[Sr35LRR] and *HvSh1*[Sr35LRR]. Representative data shown from seven replicates and scored for cell death. **d**, Western blot of hybrid receptors tested in tobacco. Pooled three replicates. Ponceau S staining as loading control. Composite image of same blot.

**e**, Cryo-EM structure of Sr35 and structural predictions of *TaSH1*[LRR] and *HvSH1*[LRR] (ref. [28]). Polymorphisms between *TaSH1* or *HvSH1*, and Sr35 are shown (orange). Residues mutated are shown red. **f**, Wheat protoplast transfections of *TaSh1*[GOF], *HvSh1*[GOF] and controls co-expressed with *AvrSr35*. Experiment and statistics as in **b**. **g**, Tobacco cell death of *TaSh1*[GOF] and *HvSh1*[GOF]. Replicates and scoring as in **c**. **h**, Western blot of GOF experiment in tobacco. Replicates and loading control as in **d**. **i**, Graphical illustration of MLA hybrid receptors; design as in **a** (*HvMla10*[Sr35LRR] and *HvMla13*[Sr35LRR]). **j**, Wheat protoplast transfections of *HvMla10*[Sr35LRR], *HvMla13*[Sr35LRR] and controls co-expressed with *AvrSr35*. Experiment and statistics as in **b**. **k**, Tobacco cell death of *HvMla10*[Sr35LRR] and *HvMla13*[Sr35LRR]. Replicates and scoring as in **c**. **l**, Western blot of MLA hybrid receptors in tobacco. Replicates and loading control as in **d**. Composite image of two independent blots.

## Engineering of CNLs for effector recognition specificity

To test whether the evolutionary conservation of CNL resistosomes can be harnessed for the design of new receptors with altered function, we first chose two closely related *Sr35* homologues (*Sh*) of unknown resistance function in bread wheat (*Triticum aestivum*; *TaSh1*) and in the sister species barley (*Hordeum vulgare*; *HvSh1*). We generated hybrid receptors of *TaSh1* and *HvSh1* in which the LRR domain, including the highly conserved WHD α4-helix, was substituted by the equivalent fragment of Sr35 (termed *TaSh1*[Sr35LRR] and *HvSh*[Sr35LRR]) (Fig. 5a and Extended Data Fig. 10a). Unlike wild-type *TaSh1* or *HvSh1* genes, both hybrid receptors mediated *AvrSr35*-dependent cell death in wheat leaf protoplasts prepared from cultivar Chinese Spring and when expressed in leaves of *N. benthamiana* (Fig. 5b–d), indicating neofunctionalization of the orphan receptors.

Owing to the high sequence similarity of *Ta*SH1 and *Hv*SH1 with *T. monococcum* Sr35 (86.5% and 86.4% amino acid sequence identity to Sr35, respectively), we reasoned that targeted amino acid substitutions in the LRR domains of the homologues might be sufficient to enable detection of AvrSr35. Combined structural model (Extended Data Fig. 10b,c) and protein sequence alignments indicated that the AvrSr35-interacting residues of Sr35 are polymorphic in *Ta*SH1 and *Hv*SH1 (Fig. 5e and Extended Data Fig. 10d). The alignments identified several residues in the LRR domains of *Ta*SH1 and *Hv*SH1 that are likely to hinder effector binding at the modelled interface. Accordingly, we generated *TaSh1* and *HvSh1* variants encoding receptors with eight and ten substitutions in the LRR, respectively (*Ta*SH1$^{D609G/Y728F/D731R/I754K/Q755R/L804W/Q810E/R857W}$ and *Hv*SH1$^{Y727F/Q801E/G754K/Q752P/Q755R/R809E/W835I/R856W/917D/P919W}$; designated for simplicity *Ta*SH1$^{GOF}$ and *Hv*SH1$^{GOF}$, respectively, in which GOF denotes gain-of-function) (Fig. 5a). Unlike wild-type *HvSh1*, *HvSh1*$^{GOF}$ mediated a clear cell death response in wheat protoplasts and *N. benthamiana* when co-expressed with the effector *AvrSr35* (Fig. 5f–h). *TaSh1*$^{GOF}$ induced a notable cell death phenotype in wheat protoplasts, but not in *N. benthamiana* (Fig. 5f–h), which is probably due to undetectable *Ta*SH1$^{GOF}$ protein in the heterologous *N. benthamiana* expression system (Fig. 5h). These findings suggest that targeted amino acid substitutions that mimic the effector binding region of Sr35 are sufficient for neofunctionalization of these orphan receptors. The relatively small number of nucleotide changes needed to enable *Ta*SH1 to detect AvrSr35 makes it feasible to introduce such changes in elite bread wheat by gene editing. In this way, generating varieties that are resistant to *Pgt* Ug99[32–36] provides an alternative strategy to transgene-mediated *Sr35* transfer from *T. monococcum* to bread wheat[5,37].

Next, we investigated whether the Sr35 LRR domain, transferred to more distant CNLs (approximately 45% amino acid sequence identity) in the sister species barley, can generate functional hybrid receptors. We chose barley *Hv*MLA10 and *Hv*MLA13, known to confer isolate-specific immunity against the barley powdery mildew fungus[38], *Blumeria graminis* f. sp. *hordei* (*Bgh*), as templates to engineer AvrSr35 recognition (Fig. 5i). The ascomycete *Bgh* effectors recognized by *Hv*MLA10 and *Hv*MLA13 lack sequence similarity to AvrSr35 from the basidiomycete *Pgt*. Using the above reasoning for hybrid receptor generation of *Sr* homologues, the LRR domains of *Hv*MLA10 and *Hv*MLA13 were replaced by the Sr35 LRR. The two resulting hybrid receptors, *HvMla10*$^{Sr35LRR}$ and *HvMla13*$^{Sr35LRR}$, induced cell death when co-expressed with *Pgt AvrSr35* in wheat protoplasts and *N. benthamiana* (Fig. 5j–l). This finding supports our hypothesis that a combination of effector binding to the LRR and steric clash of the effector with the NBD is needed for CNL activation, as exemplified here for hybrid receptors where the AvrSr35 effector is predicted to clash with the MLA NBD.

## Discussion

Our results, together with earlier data, strongly suggest that the activation and signalling mechanisms of CNLs are evolutionarily conserved. Three independent lines of evidence support this idea: (1) our structural elucidation of the wheat Sr35 resistosome and its similarity to the previously reported *Arabidopsis* ZAR1 resistosome structure[3]; (2) the functional interspecies hybrid receptors generated from the non-orthologous CNLs wheat Sr35 and barley MLAs; and (3) the conservation extends to the non-selective cation flux across membranes enabled by pentamerization, although it is possible that ion selectivity and channel dynamics differ between individual CNLs, including the channel activity of so-called helper NLRs acting downstream of canonical plant NLRs[39]. Reconstitution of effector-dependent Sr35-triggered cell death in insect cells indicates that regulated channel activity is sufficient to recapitulate plant CNL-mediated cell death in eukaryotic cells of another kingdom. It is possible that plant CNL pore formation and ion flux trigger and intersect with intrinsic cell death pathways in animals, for example, Apaf-1 apoptosome-mediated developmental

and stress-induced cell death[40,41]. Although the components needed for cell death downstream of CNL channel activity in plants remain to be identified, the evolutionary conservation of channel activity rationalizes how highly diverse pathogen signals activate a shared set of downstream responses. This is reminiscent of the highly conserved NADase activity encoded by the TIR domain, which converts the presence or activity of diverse pathogen molecules into TNL-triggered immune signals[30,31].

Our study also uncovers the mechanism by which direct or indirect recognition of pathogen effectors results in the formation of the conserved pentameric scaffold that facilitates channel activity. Indirect recognition of a bacterial pathogen effector by ZAR1 results in a conformational change of the NBD, allosterically promoting exchange of ADP with ATP/dATP for full receptor activation. Our data support a similar mechanism for Sr35 activation by direct recognition of AvrSr35. These results lend further support to the notion that the exchange of ADP with ATP/dATP is widely involved in the activation of NLRs. Although AvrSr35 is essential for the activation of Sr35, the effector makes no contribution to oligomerization of the Sr35 resistosome, which is principally mediated by the conserved NBD. This is also true for the assembly of the ZAR1 resistosome in *Arabidopsis* and the Apaf-1 apoptosome in animals[3,42]. It seems that recognition of diverse pathogen effectors by the polymorphic LRRs release the conserved NBD to mediate NLR oligomerization.

A third plant NLR recognition mechanism involves a combination of direct and indirect recognition through the incorporation of effector target domains (integrated decoys) into the NLR domain architecture, termed NLR-IDs, representing approximately 10% of all NLR-encoding genes of a plant species[43]. Crystal structures of the ID in complex with the bound pathogen effector have been resolved for two NLR-IDs, enabling structure-informed ID engineering to extend pathogen strain-specific NLR recognition[44–46]. How the corresponding full-length NLR-ID receptors are activated, including a potential steric clash with their NBD is unclear owing to a lack of full-length receptor structures. This is further complicated as NLR-IDs, which directly recognize effectors, typically function with canonical NLRs as interacting pairs[43]. Direct recognition of pathogen effectors by plant NLRs can be rapidly circumvented by polymorphisms of effector residues at the effector–NLR interface, particularly as a pathogen and its host plant typically evolve at different time scales. Virulent isolates of *Pgt* within and beyond the Ug99 lineage have escaped the recognition of at least one of the recently cloned *Sr* genes, including single amino acid changes in the effector[47]. For example, a *Pgt* isolate with combined virulence against *Sr35* and *Sr50* caused an epidemic in Sicily in 2017[48]. Our findings allow the prediction not only of AvrSr35 substitutions that might escape Sr35 recognition, but also substitutions in the Sr35 LRR that can physically 're-capture' such effector variants. More generally, the evolutionarily conserved plant CNL resistosome architecture with its conserved function highlights the future potential of structure-guided NLR engineering for crop improvement.

*Note added in proof*: After completion of this work, the Sr35 resistosome structure was confirmed in an independent study[49].

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

## Methods

### Protein expression, purification and negative staining

Codon optimized $Sr35^{L15E/L19E}$ and $AvrSr35$ genes were cloned into the pFastBac1 vector (Invitrogen) with an N-terminal 6×His-SUMO tag and an N-terminal glutathione S-transferase (GST) tag, respectively. The constructs were transformed into EMBacY[50] competent cells for recombinant bacmid DNA generation. Recombinant baculovirus was generated by initial lipofection with Xtreme gene reagent (Roche) of Sf21 insect cells (Invitrogen). Baculovirus was generally amplified to the $P_2$ generation before protein expression. $Sr35^{L15E/L19E}$ and $AvrSr35$ were co-expressed in Sf21 insect cells, 50 ml of each virus was used per 1 l of culture. After expression of protein at 28 °C for 48 h, the insect cells were ollected and resuspended with buffer containing 50 mM Tris pH 8.0, 150 mM NaCl, 0.05% Triton X-100 and 5% glycerol. The cell lysates generated by sonication were centrifuged at 13,000 r.p.m. for 1.5 h, and then the supernatant was collected. The protein complex was purified with glutathione Sepharose 4B (GS4B) resin. After binding to the glutathione agarose twice, the agarose was washed with three column volumes of resuspension buffer, and the tagged protein complex was treated with GST-tagged PreScission protease at 4 °C overnight to remove GST and 6xHis-SUMO tags simultaneously. The digested protein complex in the flow-through was concentrated and subjected to HiLoad superpose 6 column (GE) in buffer containing 50 mM Tris pH 8.0, 100 mM NaCl and 0.01% Triton X-100. Pooled peak fractions were used for cryo-EM sample preparation.

### Cryo-EM sample preparation and data collection

The Sr35–AvrSr35 complex grids were prepared for cryo-EM analysis. Holy carbon grids (Quantifoil Au 1.2/1.3, 300 mesh) were glow-discharged for 30 s at medium level in HarrickPlasma after 2 min evacuation. The purified Sr35–AvrSr35 protein was concentrated to approximately 0.5 mg ml$^{-1}$ and 3 μl of sample were applied to the grid. The grids were blotted for 2–3 s using a pair of filter papers (55 mm, Ted Pella Inc.) at 8 °C with 100% humidity and flash-frozen in liquid ethane using a FEI Vitrobot Marked IV. Stacks of Sr35–AvrSr35 cryo-EM samples were collected by a Titan Krios microscope operated at 300 kV, equipped with a K3 Summit direct electron detection camera (Gatan) using EPU 2 (Thermo Fisher Scientific, 2.8.1.10REL) at Zhengzhou University. Micrographs were recorded at 81,000× magnification corresponding to 1.1 Å per pixel. The defocus ranged from −1.5 μm to −2.0 μm. Each image stack contains 32 frames recorded every 0.11 s for an accumulated dose of approximately 50 e− per Å$^2$ and a total exposure time of 3.5 s. A second dataset from an independent protein purification was recorded at EMBL Heidelberg with the following parameters: Titan Krios microscope operated at 300 kV, equipped with a K3 Summit direct electron detection camera (Gatan), 50 e per Å$^2$, 40 frames per stack.

### Image processing and 3D reconstruction

All micrographs of the Sr35–AvrSr35 complex were 2 × 2 binned, generating a pixel size of 1.1 Å. The MotionCor2 program was used to perform Motion correction[51]. Contrast transfer function (CTF) parameters were estimated by CTFFIND4[52]. On the basis of the CTF estimations, 5,292 micrographs were manually picked and were further processed in RELION3.1[53].

1,608,441 particles were picked using Laplacian-of-Gaussian auto picking and then subjected to several rounds of 2D classification[54,55]. Every round of 2D classification performed 25 iterations with regularisation parameter $T = 2$ and number of classes = 100 to remove bad particles. The particles with the best quality were used to generate the initial model using ab initio calculation from RELION3.1. Then 698,386 particles were imported into 3D classification with $C1$ symmetry. There were five Sr35 molecules in the complex, each of which was bound to one AvrSr35 molecule. $C5$ symmetry was used in the following 3D refinement. After CTF refinement and postprocessing, the resolution of the Sr35–AvrSr35 complex reconstruction was 3.0 Å. The resolution was estimated by the gold-standard Fourier shell correlation = 0.143 criterion[56]. Local resolution distribution was evaluated using RELION 3.1 (ref. [57]).

In the reconstruction above, the LRR and AvrSr35 portions were more flexible than the other parts of the Sr35–AvrSr35 complex. To improve the density of the more flexible portions, we used a procedure as previously described[58]. The final refined particles were expanded with $C5$ symmetry. A local mask was generated using USCF Chimera[59]. Expanded particles and local mask were subjected to 3D classification without alignment. Finally, 476,069 particles were used for 3D auto-refinement and CTF refinement. A final resolution of 3.33 Å was achieved after postprocessing. For the second dataset, one third of the micrographs were analysed the same way and resulted in the same overall structure at a resolution of 3.4 Å. The resulting model was not used further for model building.

### Model building and refinement

The final density map was obtained by merging the global map and the local map which contained LRR and AvrSr35, using a 'combine_focused_map' in PHENIX 1.18.2 (ref. [60]). The model of the Sr35–AvrSr35 complex was manually built in COOT 0.9 (ref. [61]) based on the global and the local maps. The generated model was refined against the combined Sr35–AvrSr35 EM density using real space refinement in PHENIX with secondary structure and geometry restraints[61]. Model statistics can be found in Extended Data Table 1. USCF Chimera 1.15 and ChimeraX 1.15 were used to visualize models and density maps.

### Transient gene expression assays in wheat protoplasts

Seedlings of the wheat cultivar. Chinese Spring were grown at 19 °C, 70% humidity and under a 16 h photoperiod. Protoplasts were isolated from the leaves and transfected as previously described[23]. The coding sequences of $TaSh1$ (NCBI XP_044359492.1) and $HvSh1$ (NCBI KAE8803279.1) were generated by gene synthesis based on wild-type codons (GeneArt, Invitrogen). The coding sequence of all tested receptor constructs, or an EV as negative control, were expressed from $pIPKb002$ vector[62] containing the strong ubiquitin promoter. Receptors were co-expressed with $AvrSr35$ in $pIPKb002$. In addition, cotransfection of $pZmUBQ:LUC$[63] facilitated the expression of the $LUC$ reporter construct. Each treatment was transfected with 4.5 μg of $pZmUBQ:LUC$ and 5 μg of $pIPKb002:AvrSr35$. Quantities of receptor-encoding $pIPKb002$ plasmid were varied for each construct in an effort to minimize cell death due to (receptor) toxicity-mediated cell death ($EV$ 8 μg; $Sr35$ and $Sr35$ mutants 2 μg; $AvrSr35$ and $AvrSr35$ mutants 5 ug; $HvMla10$, $HvMla13$, $HvMla10^{Sr35LRR}$, $HvMla13^{Sr35LRR}$, $TaSh1$, $TaSh1$, $TaSh1^{GOF}$, $TaSh1^{GOF}$ 8 μg; $TaSh1^{Sr35LRR}$ and $TaSh1^{Sr35LRR}$ 2 μg). A maximum of two technical replicates were completed with the same batch of wheat seedlings. Luminescence was measured using a luminometer (Centro, LB960). Relative luminescence was calculated by dividing the absolute luminescence value by that of the corresponding $EV$ treatment (EV = 1).

### Transient gene expression and western blotting in tobacco

For $N. benthamiana$ transient gene expression, $Sr35$ and $Sr35$ mutants, $AvrSr35$ and $AvrSr35$ mutants were cloned into the $pDONR$ vector (Invitrogen). The obtained plasmids of $Sr35$ and $Sr35$ mutants were recombined by an LR clonase II (Thermo Fisher Scientific) reaction into $pGWB517$-4×Myc with a C-terminally fused 4×Myc epitope tag[64], while $AvrSr35$ and $AvrSr35$ mutants were recombined into the $pXCSG-mYFP$[65] vector with a C-terminally fused mYFP epitope tag. After being verified by Sanger sequencing, all the constructs were transformed into $Agrobacterium tumefaciens$ GV3101 pMP90RK by electroporation. Transformants were grown on LB media selection plates containing rifampicin (15 mg ml$^{-1}$), gentamycin (25 mg ml$^{-1}$), kanamycin (50 mg ml$^{-1}$), and spectinomycin (50 mg ml$^{-1}$) for transformants harbouring $pGWB517$-4×Myc or carbenicillin (50 mg ml$^{-1}$) for $pXCSG-mYFP$.

Individual *Agrobacterium* transformants were picked and cultured in LB medium containing respective antibiotics in the abovementioned concentration. After shaking culture at 28 °C for 16 h, the culture was harvested at 3,800 r.p.m. for 10 min and resuspended with infiltration buffer containing 10 mM MES pH 5.6, 10 mM MgCl$_2$ and 150 μM acetosyringone. The OD$_{600}$ of *AvrSr35* and *AvrSr35* mutant strains was adjusted to 1.0. For *Sr35* and *Sr35* substitution mutants, the OD$_{600}$ was adjusted to 0.15. Hybrid receptor bacterial strains (*HvMla10$^{Sr35LRR}$*, *HvMla13$^{Sr35LRR}$*, *TaSh$^{Sr35LRR}$*, *HvSh$^{Sr35LRR}$*) were adjusted to an OD$_{600}$ of 0.6. In the hybrid receptor gain-of-function experiment, the OD$_{600}$ of *TaSh1*, *HvSh1*, *TaSh1$^{GOF}$* and *HvSh1$^{GOF}$* bacterial strains was adjusted to 1.8 without resulting in cell death in co-expression of *TaSh1* and *HvSh1* when co-expressed with *AvrSr35*. After dilution, all the cell suspensions were incubated at 28 °C for 1 h at 200 rpm. Construct expression was conducted in leaves of four-week-old *N. benthamiana* plants via *Agrobacterium*-mediated transient expression assays. For phenotypic experiments, *Agrobacteria* cultures expressing receptor constructs, or the respective receptor mutants, were co-infiltrated with *AvrSr35*, or its mutants, at 1:1 ratio using a syringe. As a control, either receptor or effector bacterial strains were replaced with *Agrobacteria* transformed with EVs. Phenotypic data were recorded at day 3 after infiltration.

*Agrobacterium*-mediated transient expression assays for protein detection were conducted as described above. The infiltrated leaves were harvested at 24 h after infiltration, flash-frozen in liquid nitrogen and ground to powder using a Retsch grinder. Plant powder was mixed with 4xLämmli buffer in a 1:2 ratio. Five microlitres was loaded onto 10% SDS–PAGE. After transfer to PVDF membrane, protein was detected using monoclonal mouse anti-MYC (1:3,000; R950-25, Thermofisher), polyclonal rabbit anti-GFP (1:3,000; pabg1, Chromotek), polyclonal goat anti-mouse IgG-HRP (1:7,500; ab6728, Abcam) and polyclonal swine anti-rabbit IgG-HRP (1:5,000; PO399, Agilent DAKO) antibodies. Protein was detected using SuperSignal West Femto:SuperSignal substrates (ThermoFisher Scientific) in a 1:1 ratio.

### Electrophysiology

The TEVC recordings were conducted as previously described[4]. The cDNAs of *Sr35*, or *Sr35* mutants, and *AvrSr35* were cloned into the *pGHME2* plasmid for expression in *Xenopus* oocytes. cRNAs for all constructs were transcribed using T7 polymerase. Ovarian lobes were obtained from adult *Xenopus laevis* under anaesthesia. Both the amount of cRNA injected and the oocyte incubation time were optimized to minimize toxicity caused by the assembled Sr35 resistosome. Isolated oocytes were co-injected with 0.5 ng cRNA of *Sr35* (WT and mutants) and *AvrSr35*. Oocytes were then incubated at 18 °C for approximately 4 h in ND96 buffer (96 mM NaCl, 2.5 mM KCl, 1 mM MgCl$_2$, 1.8 mM CaCl$_2$, 5 mM HEPES pH 7.6) supplemented with 10 μg l$^{-1}$ penicillin and 10 μg l$^{-1}$ streptomycin. TEVC measurements were performed between 4–7 h later after injection. Water-injected oocytes served as controls.

Two-electrode voltage-clamp recordings were performed using an OC-725C oocyte clamp amplifier (Warner Instruments) and a Digidata 1550 B low-noise data acquisition system with pClamp 10.6 software (Molecular Devices). Data were analysed using OriginPro, 2022 (OriginLab). The microelectrode solutions contained 3 M KCl (electrical resistance of 0.5–1 MΩ), and the bath electrode was a 3 M KCl agar bridge. To eliminate the chloride currents mediated by endogenous Ca$^{2+}$-activated chloride channels in Xenopus oocytes, the ND96 recording solution was supplemented with 200 μM CaCC inhibitor (CaCCinh)-A01, and the oocytes were pre-incubated 5–10 min before measurement. To test the channel blocking effect of LaCl$_3$, the oocytes were pre-incubated for 5–10 min in the recording solutions supplemented with 200 μM CaCCinh-A01 and 100 μM LaCl$_3$ before measurement. For the recordings in Fig. 3g, the various recording solutions were as follows: KCl (96 mM), K-gluconate (96 mM), NaCl (96 mM), Na-gluconate (96 mM) and TBA-Cl (96 mM). All solutions contained 5 mM HEPES pH 7.6, and 1 mM MgCl$_2$ or Mg-gluconate. For the recordings in Fig. 3h, the various recording solutions were as follows: CaCl$_2$ (12 mM), Ca-gluconate (12 mM), MgCl$_2$ (12 mM) and Mg-gluconate (12 mM). All solutions contain 5 mM HEPES pH 7.6, and 1 mM MgCl$_2$ or Mg-gluconate. The treatments of CaCCinh-A01 and LaCl$_3$ were conducted as above. Voltage-clamp currents were measured in response to voltage steps lasting 7.5 s and to test potentials ranging from −110 mV to +70 mV, in 20 mV increments. Before each voltage step, the membrane was held at 0 mV for 1.60 s, and following each voltage step, the membrane was returned to 0 mV for 2 s. *I–V* relations for Sr35 resistosome channels were generated from currents that were measured 0.2 s by the end of each test voltage step. Three independent batches of oocytes were investigated and showed consistent findings. Data from one representative oocyte batch are shown.

### Statistics and reproducibility

No statistical method was used to predetermine sample size. Sample size was chosen in accordance with the generally accepted standards of the resprective scientific field. Data distribution for each protoplast transfection experiment was subjected to the Shapiro-Wilk normality test. All experiments were found to be normally distributed. An ANOVA and subsequent Tukey post hoc test was completed for each experiment. Treatments found to be significantly different were labelled with different letters ($\alpha$ = 0.05). All statistical output is listed in Supplementary Information.

Purification of the Sr35 resistosome was performed more than 10 times. Pull-down and SDS analysis were highly reproducible between biological replicates and comparable with Extended Data Fig. 1b,c. Negative staining was performed for each protein preparation and showed some variability compared to Fig. 1a, but generally yielded >20% star-shaped particles. Cryo-EM datasets were recorded twice from independent protein preparations (micrograph of one cryo-EM sample preparation shown in Extended Data Fig. 1d) and yielded highly similar cryo-EM density maps.

Insect cell death data were performed with six biological replicates and yielded comparable results to Extended Data Fig. 1a.

Tobacco agroinfiltration data was performed with at least two biological replicates for each substitution mutant and always simultaneously with western blot analysis. Technical replicates of one dataset are shown as raw image data. Western blot samples were always obtained from the same biological replicate as the phenotypic data. Only phenotypic data for which the western blot gave a clear signal are shown.

### Ethics declarations

The animal study (*Xenopus laevis*) was reviewed and approved by the Laboratory Animal Ethics Committee at Institute of Genetics and Developmental Biology, Chinese Academy of Sciences, Beijing, China with the approval ID AP2020029.

### Reporting summary

Further information on research design is available in the Nature Research Reporting Summary linked to this article.

## Data availability

The atomic coordinates of the Sr35 resistosome have been deposited in the Protein Data Bank (PDB) with the accession code 7XC2. The EM map for the local mask of Sr35 LRR in complex with AvrSr35 has been deposited in the Electron Microscopy Data Bank (EMDB) with the accession code EMD-33111. Sequences of *TaSh1* and *HvSh1* are available at NCBI under accession codes XP_044359492.1 and KAE8803279.1, respectively. Source data of tobacco agroinfiltrations, western blots, insect cell viability and wheat protoplast cell death are provided with this manuscript. All plasmids are available from the authors.

## Code availability

No custom codes were generated for this study.

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

**Acknowledgements** We acknowledge Zhengzhou University and EMBL Heidelberg for their assistance with cryo-EM data acquisition. We thank U. Neumann (Max Planck Institute for Plant Breeding Research, Cologne, MPIPZ) and F. Babatz (CECAD, Cologne) for TEM support, J. Jirschitzka for technical support (MPIPZ) and N. Donnelly (MPIPZ) for manuscript polishing. We thank our funding bodies: the Alexander von Humboldt Foundation (a Humboldt professorship to J. Chai), the Max-Planck-Gesellschaft (P.S.-L. and a Max Planck fellowship to J. Chai), Deutsche Forschungsgemeinschaft SFB-1403-414786233 (J. Chai and P.S.-L.) and Germany's Excellence Strategy CEPLAS (EXC-2048/1, Project 390686111) (J. Chai and P.S.-L.), iNEXT-Discovery for funding Cryo-EM dataset collection at EMBL Heidelberg (PID 16414 to A.F. and J. Chai), the National Key Research and Development Program of China 2021YFA1300701 (Z.H.), the Strategic Priority Research Program of the Chinese Academy of Sciences (XDA24020305 to Y.C.) and the National Key Research and Development Program of China (2020YFA0509903 to Y.C.).

**Author contributions** A.F. and J. Chai conceived the study. A.F., Y.C., P.S.-L. and J. Chai conceptualized the study. A.F., Y.C., P.S.-L. and J. Chai acquired funding. A.F., E. Li, A.W.L., Y. D., and J. Chang designed the experiments. A.F. and E. Li performed the biochemistry; and A.F., E.Li and Y.S. acquired the structure. J.W. and X.Z. assisted during structure study. A.F., E. Li, A.W.L. and E. Logemann performed plant functional studies. Y.D. performed electrophysiology study. A.F., E. Li, A.W.L., Y.D., Z.H. and Y.C. analysed the data. A.F., P.S.-L., J. Chai and Y.C. wrote the manuscript with input from all authors.

**Funding** Open access funding provided by Max Planck Society.

**Competing interests** The authors declare no competing interests.

**Additional information**
**Correspondence and requests for materials** should be addressed to Yuhang Chen, Paul Schulze-Lefert or Jijie Chai.

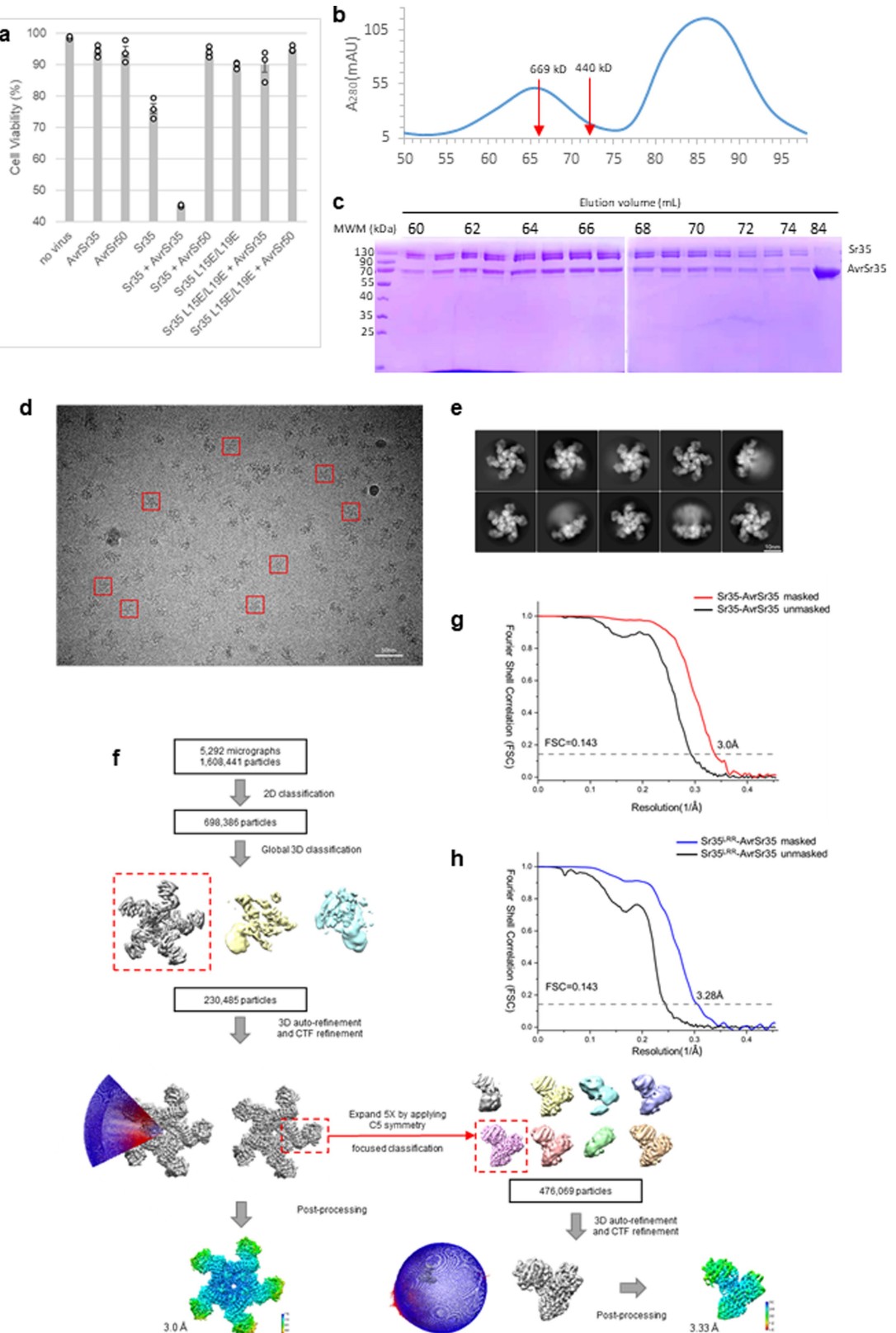

**Extended Data Fig. 1 | Sr35-AvrSr35 complex reconstitution in Sf21 insect cells. a**, Cell viability data in Sf21 insect cells. Sr35 constructs carrying amino-terminal 6xHis-Sumo-tag and Avr35 constructs carrying N-terminal GST-tag. Cell viability was determined using trypan blue stain (mean ± SEM; n = 3 technical replicates). Six biological replicates were performed with comparable results. **b**, Chromatogram of Sr35-AvrSr35 resistosome purification using HiLoad S6 column. Red arrows corresponding to 669 kDa (66 mL) thyroglobulin molecular weight marker, 440 kDa (72 mL) to ferritin. **c**, SDS–PAGE of individual fractions collected in (b). Numbers represent elution volumes. Molecular weight marker (MWM) on left. **d**, Representative cryo-EM micrograph of Sr35-AvrSr35 complex. **e**, Representative 2D class averages of Sr35-AvrSr35 complex. **f**, Flowchart of cryo-EM data processing and Sr35-AvrSr35 3D reconstruction. **g**, FSC curves at 0.143 of the final model of Sr35-AvrSr35 complex. **h**, FSC curves at 0.143 of the final model of Sr35LRR-AvrSr35.

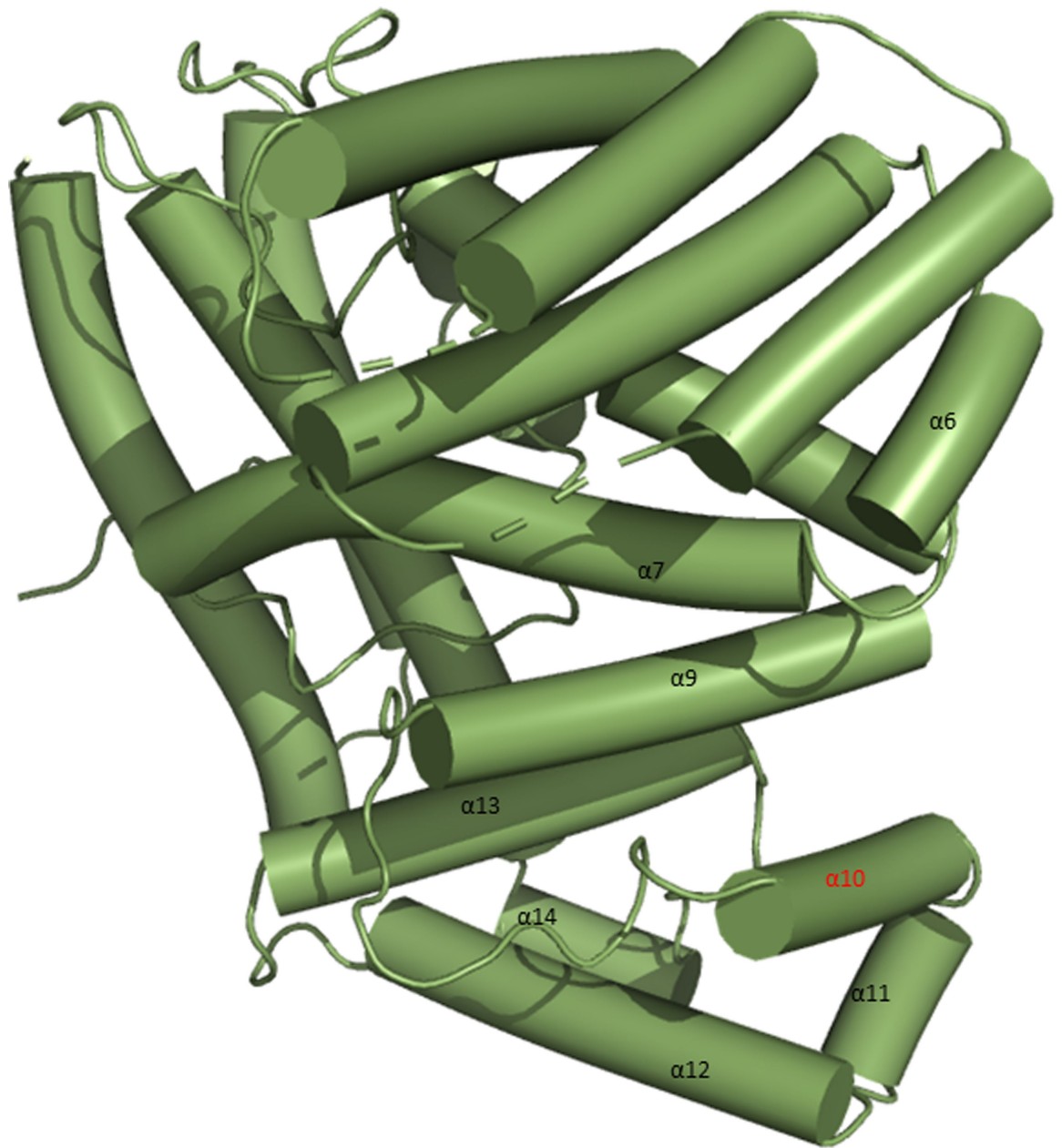

**Extended Data Fig. 2 | AvrSr35 structure from the Sr35 resistosome.** α10-helix (red) is involved in most extensive contacts with Sr35 LRR.

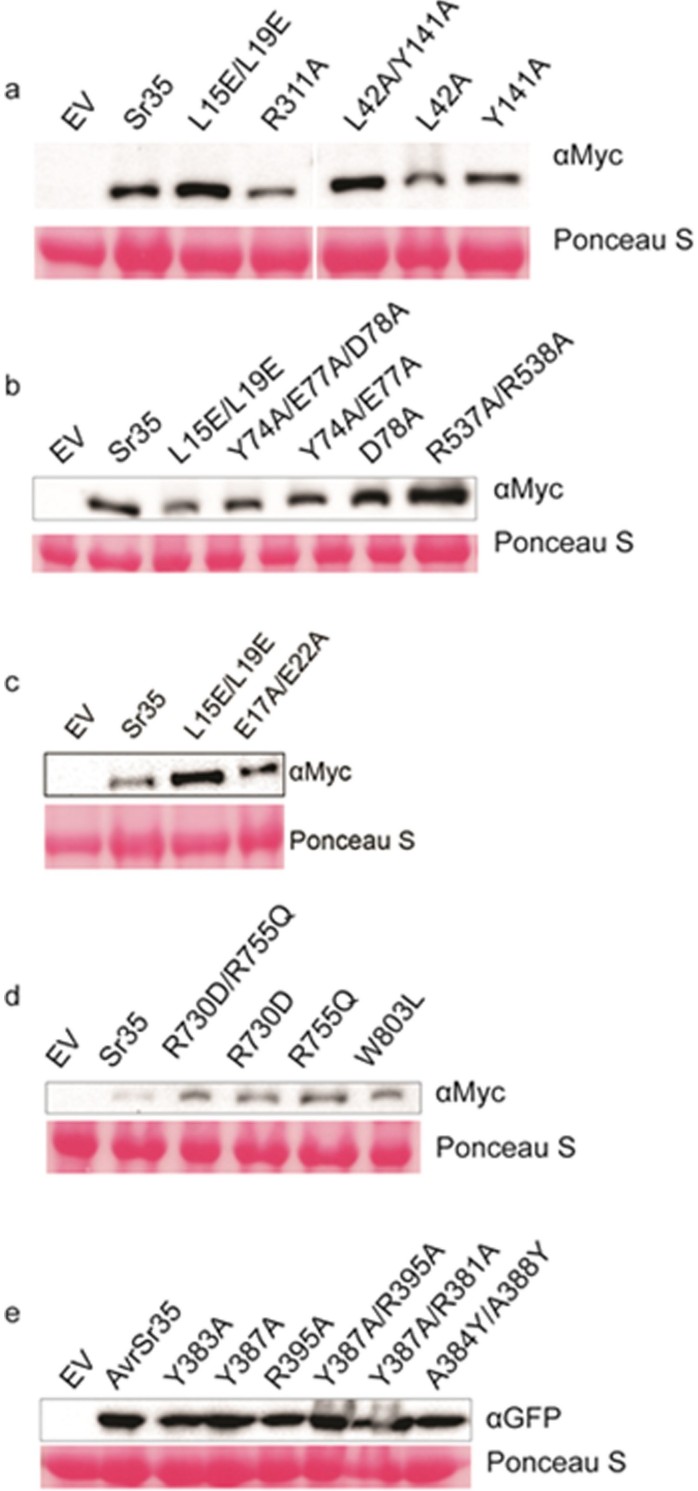

**Extended Data Fig. 3 | Western blot of *N. benthamiana* experiments.** Pooled samples from 3 technical replicates. Ponceau S staining as a loading control. **a**, Sr35 NBD ATP-binding and coiled-coil protomer interface mutants. Myc-tagged protein. Left and right side merged from the same blot. **b**, Sr35 EDVID and arginine-cluster mutants. Myc-tagged protein. Last lane cropped from the same blot. **c**, Sr35 channel mutants. Myc-tagged protein. **d**, Sr35 LRR mutants. Myc-tagged protein. **e**, AvrSr35 mutants. YFP-tagged protein detected by GFP antibody.

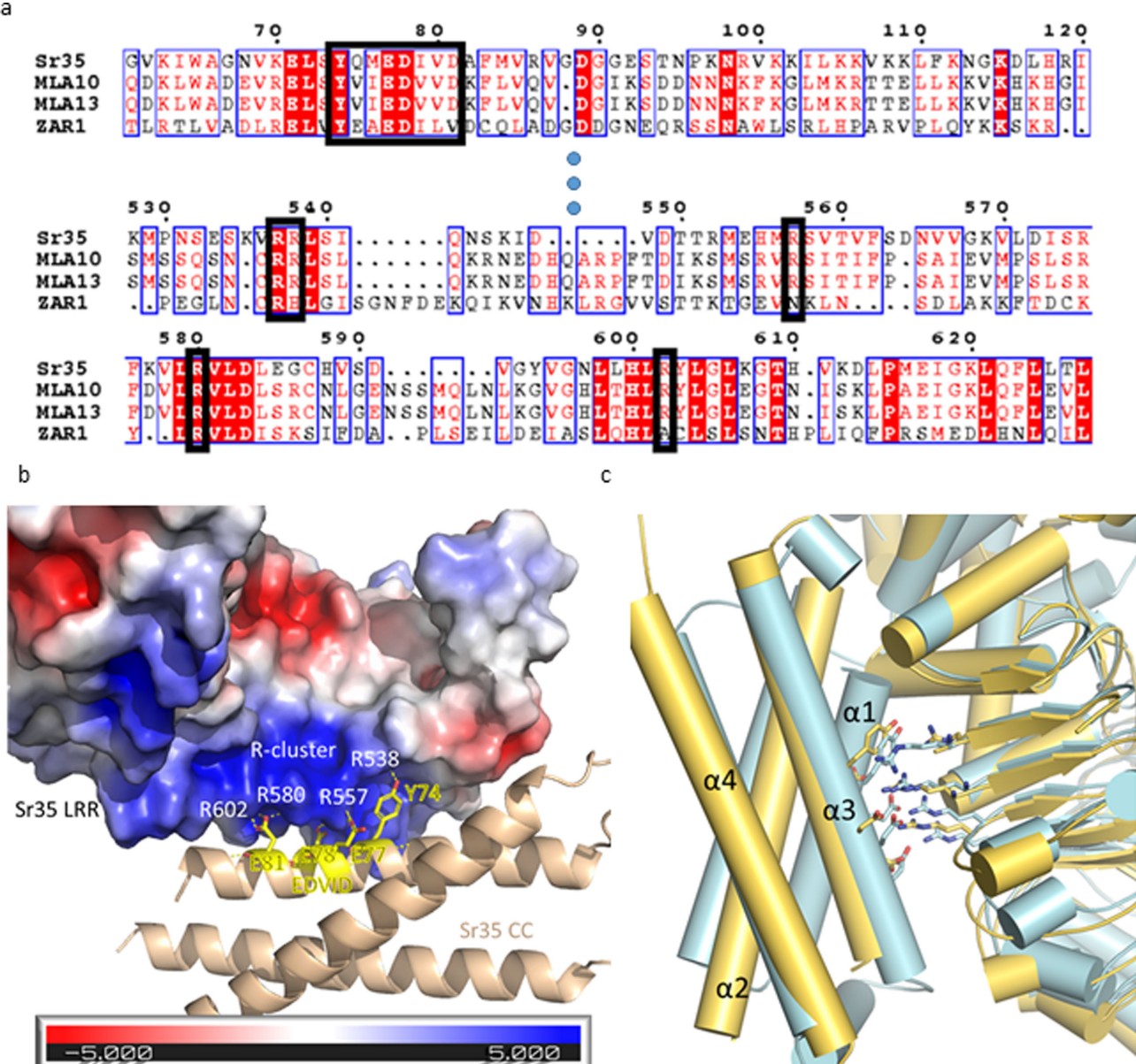

**Extended Data Fig. 4 | Details of EDIVD and R-cluster. a**, Multiple protein sequence alignment of *Hv*MLA10, *Hv*MLA13, Sr35 and ZAR1. Amino acids highlighted in red and in red text are identical and possess similar properties, respectively. Alignment of the EDIVD motif and arginine cluster are boxed in black (Robert and Gouet 2014). **b**, Electrostatic surface charge of Sr35 LRR around the EDIVD motif. **c**, Structural alignment of Sr35 inactive structure prediction (cyan) and one protomer (yellow) from Sr35 resistosome. Detailed view of EDIVD and arginine cluster interactions. In analogy to ZAR1, the Sr35 coiled-coil (CC) α1-helix might undergo structural rearrangement, which likely requires EDVID with arginine cluster interactions to transiently resolve.

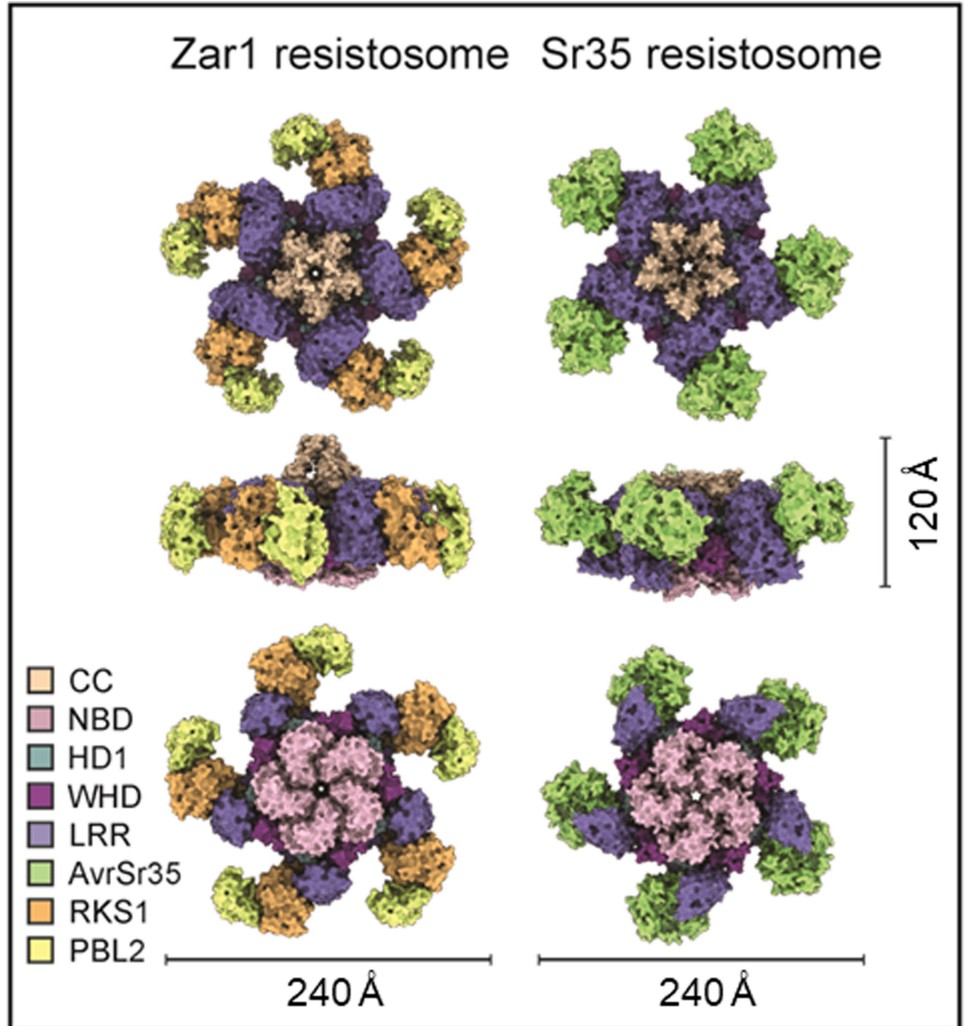

**Extended Data Fig. 5 | CNL resistosome structural conservation.**
The structures (in surface representation) of the ZAR1 resistosome and the Sr35 resistosome are shown. Zar1 is indirectly activated by the host proteins PBL2 and RKS1. Sr35 is directly activated by the fungal effector AvrSr35. The first, second, and third row show the top, side, and bottom views of these structures, respectively. Domains are coloured according to in-figure legend. Sizes are indicated by scale bar.

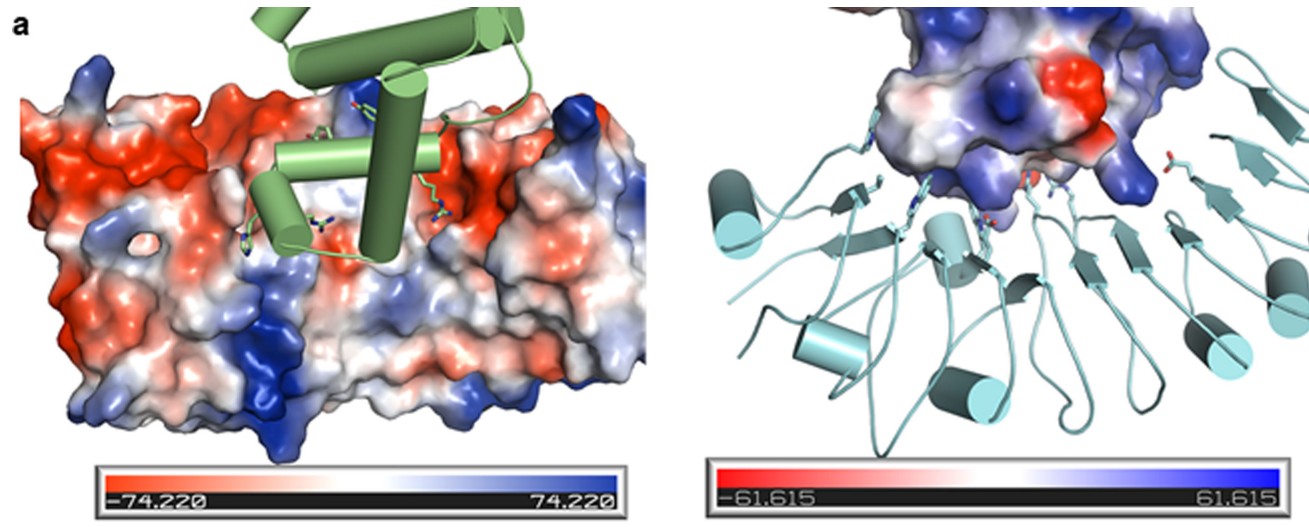

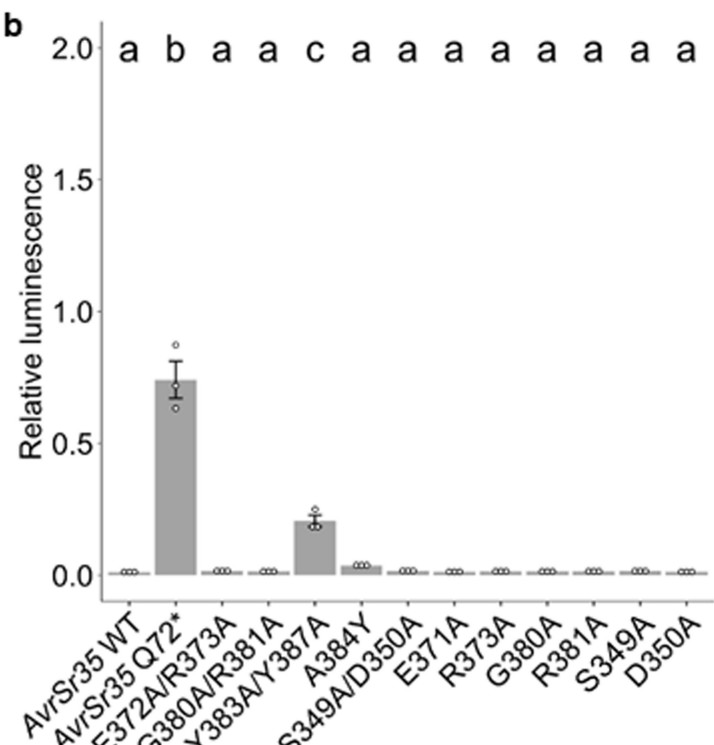

**Extended Data Fig 6 | Recognition of AvrSr35 effector by Sr35 LRR domain.**
**a**, Shape and charge complementarity of Sr35 LRR and AvrSr35 at their interface. (Left) AvSr35 shown as cartoon (lime) and Sr35 as electrostatics surface model. (Right) Sr35 LRR shown as cartoon (cyan) and AvrSr35 as electrostatics surface model. **b**, Wheat protoplast data of AvrSr35 mutants predicted to impair Sr35 recognition. Relative luminescence as readout for cell death. Empty vector treatment defined the relative baseline (mean ± SEM; $n$ = 3). Test statistics derived from ANOVA and Tukey post hoc tests ($P$<0.05). Exact $p$ values provided in Supplementary Table 3.

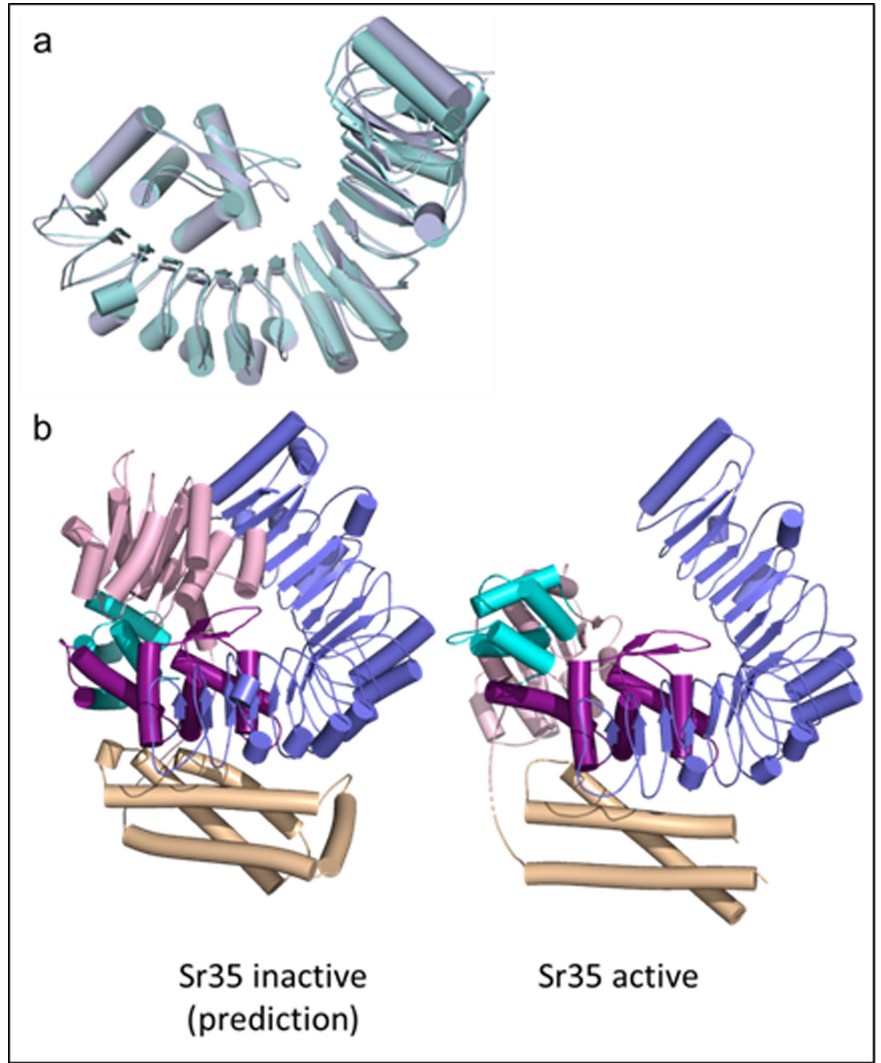

**Extended Data Fig. 7 | Comparison of the Sr35 prediction (AlphaFold2) with the Sr35 protomer from the cryo-EM structure. a**, Structural alignment of WHD and LRR domains from Sr35 AlphaFold2 prediction (cyan) and from Sr35 resistosome Cryo-EM structure (blue). **b**, Structural comparison of monomeric Sr35 from prediction (left) and from Cryo-EM structure (right). Substantial differences exist highlighting the structural re-organization within the NOD module (NBD-HD1 relative to WHD). Domain color code: coiled-coil (yellow), NBD (light pink), HD1 (cyan), WHD (purple), and LRR (blue).

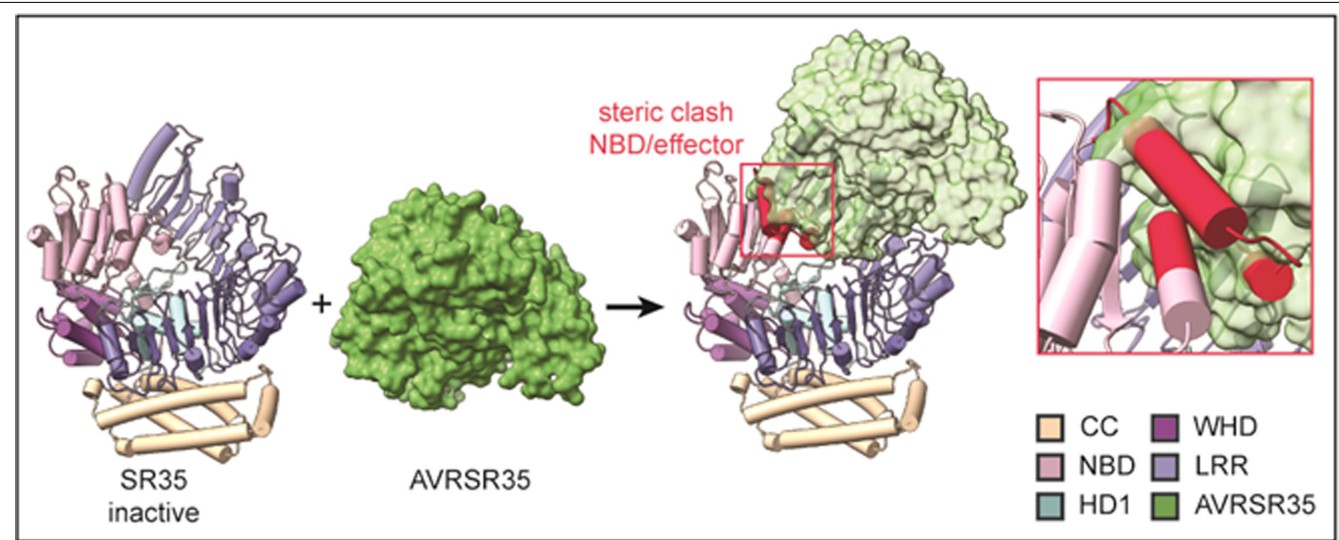

**Extended Data Fig. 8 | Steric clash between AvrSr35 and Sr35 NBD mediates Sr35 receptor activation.** Inactive Sr35 inside the cell comes in contact with Pgt effector AvrSr35. In avoidance of a steric clash (red) between AvrSr35 and the Sr35 NBD domain, the Sr35 NBD domain is forced to structurally rearrange and a 'primed' receptor-effector complex is formed. Full activation and oligomerization requires subsequent ADP release, ATP binding and, NOD module rearrangement and coiled-coil (CC) domain structural rearrangement. Sr35 domains and AvrSr35 are coloured according to in-figure legend.

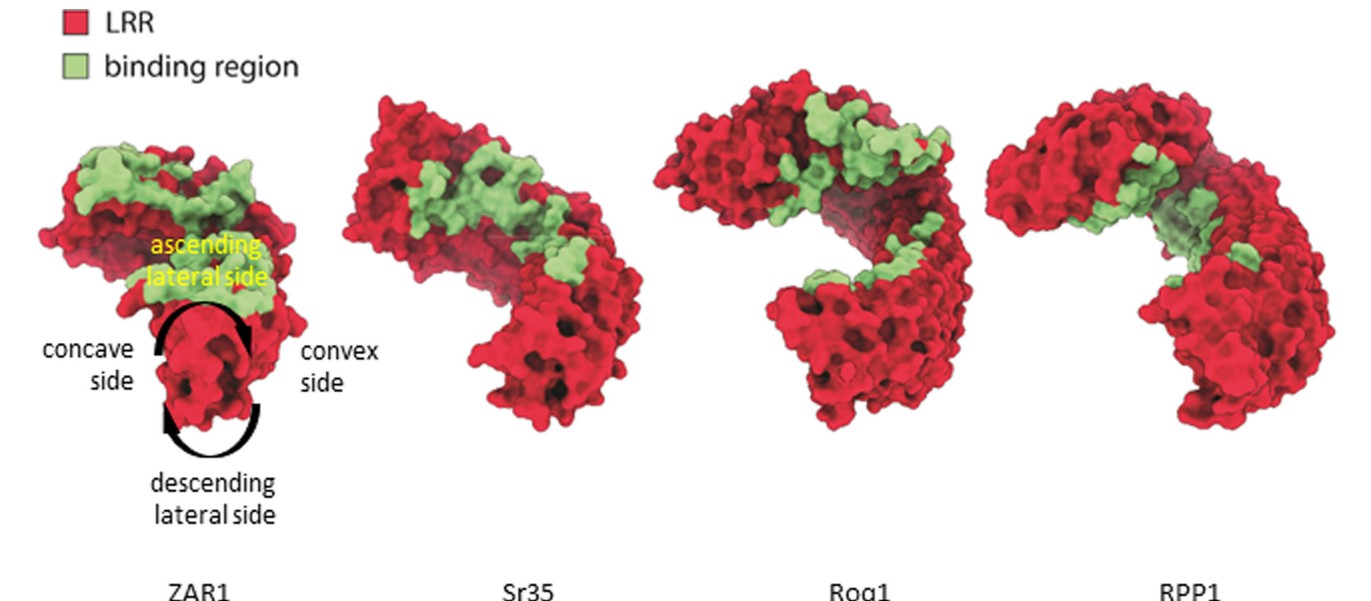

**Extended Data Fig. 9 | Comparison of ZAR1, Sr35, ROQ1, RPP1 ligand binding sites.** Ligand binding to LRR of CNLs (Zar1, Sr35) and LRR-CJID of TNLs (Roq1, RPP1) occurs in equivalent region in the ascending lateral side of the LRR domain (compare concave, convex, ascending and descending lateral sides defined on Zar1).

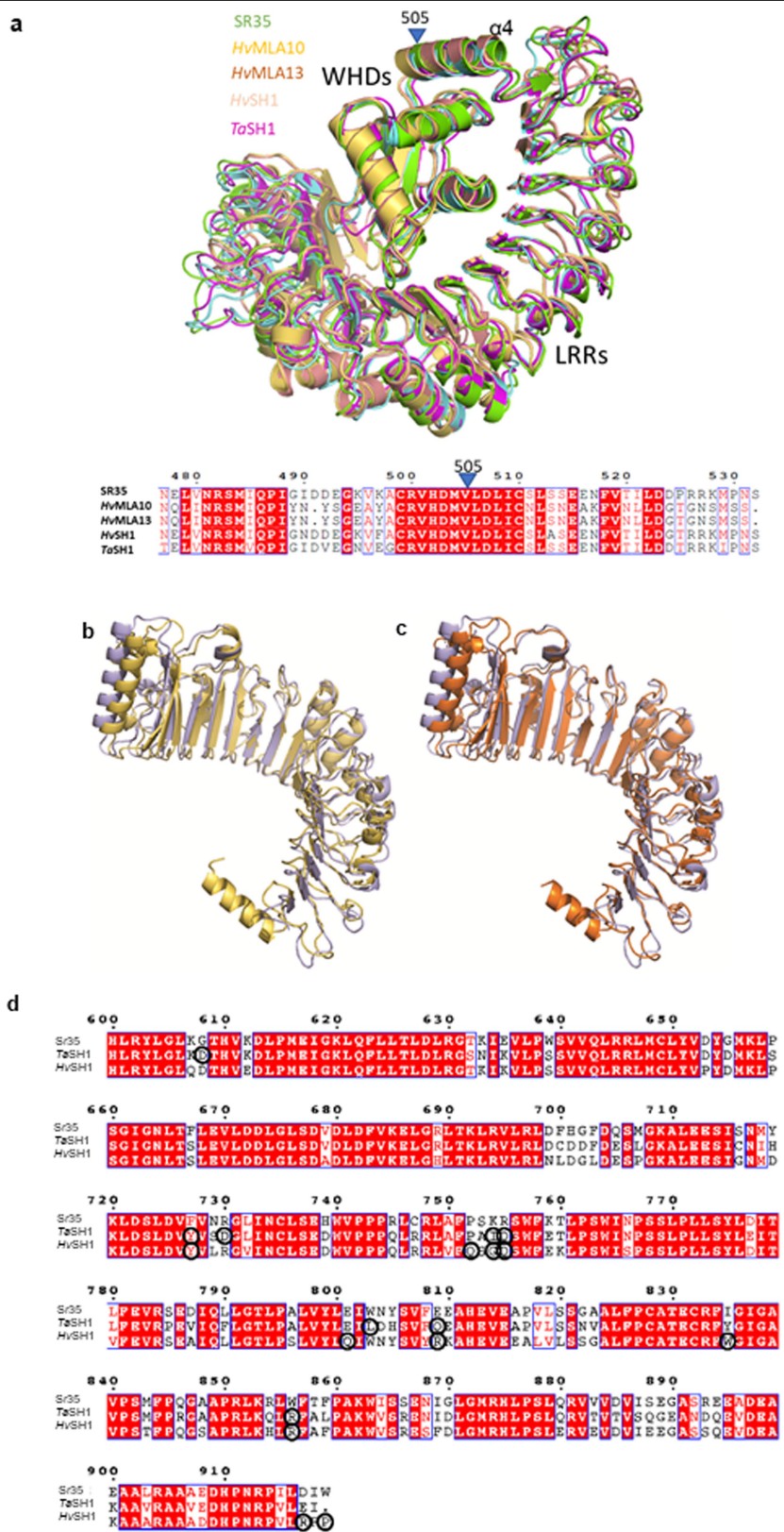

**Extended Data Fig. 10 | Rationale for hybrid receptor design. a**, Structural (top) and sequence (bottom) alignment of Sr35, HvMLA10, HvMLA13, TaSH1 and HvSH1. Amino acid 505 in the structurally and sequence conserved α4-helix of the WHD of Sr35 was included in hybrid CNL receptors. Structure of Sr35 is isolated from the cryo-EM Sr35 resistosome structure, while HvMLA10, HvMLA13, TaSH1 and HvSH1 were predicted using AlphaFold2. **b**, Structural alignment of Sr35 LRR (light blue) with structural prediction of TaSh1 (yellow) and **c**, HvSh1 (orange). **d**, Multiple protein sequence alignment of Sr35, TaSH1 and HvSH1. Circled amino acids were substituted to corresponding amino acids in the Sr35 sequence for the generation of TaSh1$^{GOF}$ and HvSh1$^{GOF}$ constructs. Amino acids highlighted in red and in red text are identical and possess similar properties, respectively.

**Extended Data Table 1 | Cryo-EM data collection, refinement and validation statistics**

| | Sr35-AvrSr35 | Sr35$^{LRR}$-AvrSr35 |
|---|---|---|
| PDB and EMDB ID | 7XC2 EMD-33112 | EMD-33111 |
| **Data collection** | | |
| Cryo electron microscope | FEI Titan Krios | |
| Voltage (kV) | 300 | |
| Detector | Gatan K3 Summit | |
| Energy filter slit width (eV) | 20 | |
| Magnification | 81,000x | |
| Pixel size (Å) | 1.1 | |
| Total electron exposure (e⁻/Å²) | 50 | |
| Number of frames collected | 32 | |
| Defocus range (mm) | -1.5 ~ -2.0 | |
| Automation software | EPU2 | |
| Micrographs collected | 5,292 | |
| Micrographs used | 5,292 | |
| **3D reconstruction** | | |
| Software | RELION 3.1 | RELION 3.1 |
| Total extraced particles | 1,608,441 | 1,152,425 |
| Total number particles for final refinement | 230,485 | 476,069 |
| Symmetry imposed | C5 | C1 |
| Resolution range (Å) | 2.8~5.7 | 3.25~4.66 |
| Resolution (Å) after refinement (FSC=0.143) | 3.41 | 4.00 |
| Resolution (Å) after post-processing (FSC=0.143) | 3.00 | 3.33 |
| Map sharpening B-factor (Å²) | -60 | -60 |
| **Refinement and validation** | | |
| Software | Phenix.real_space_refine | None |
| Model resolution (Å) | 3.0 (FSC=0.5) | |
| Model composition | | |
| Non-hydrogen atoms | 52070 | |
| Protein residues | 6445 | |
| Map-model CC (overall/local) | 0.89/0.88 | |
| B factors (Å²) | 134 | |
| R.M.S deviations | | |
| Bonds lengths (Å) | 0.003 | |
| Bonds angles (°) | 0.580 | |
| MolProbity score | 1.76 | |
| Clash score | 6.54 | |
| Rotamer outliers (%) | 1.1 | |
| Cb outliers (%) | 0.00 | |
| CaBLAM outliers (%) | 2.47 | |
| EMRinger score | 2.27 | |
| Ramachandran plot statistics | | |
| Preferred (%) | 94.65 | |
| Allowed (%) | 5.35 | |
| Outlier (%) | 0.00 | |

# Reporting Summary

## Statistics

For all statistical analyses, confirm that the following items are present in the figure legend, table legend, main text, or Methods section.

| n/a | Confirmed | |
|---|---|---|
| ☐ | ☒ | The exact sample size (*n*) for each experimental group/condition, given as a discrete number and unit of measurement |
| ☐ | ☒ | A statement on whether measurements were taken from distinct samples or whether the same sample was measured repeatedly |
| ☐ | ☒ | The statistical test(s) used AND whether they are one- or two-sided<br>*Only common tests should be described solely by name; describe more complex techniques in the Methods section.* |
| ☐ | ☒ | A description of all covariates tested |
| ☐ | ☒ | A description of any assumptions or corrections, such as tests of normality and adjustment for multiple comparisons |
| ☐ | ☒ | A full description of the statistical parameters including central tendency (e.g. means) or other basic estimates (e.g. regression coefficient) AND variation (e.g. standard deviation) or associated estimates of uncertainty (e.g. confidence intervals) |
| ☐ | ☒ | For null hypothesis testing, the test statistic (e.g. *F*, *t*, *r*) with confidence intervals, effect sizes, degrees of freedom and *P* value noted<br>*Give P values as exact values whenever suitable.* |
| ☒ | ☐ | For Bayesian analysis, information on the choice of priors and Markov chain Monte Carlo settings |
| ☒ | ☐ | For hierarchical and complex designs, identification of the appropriate level for tests and full reporting of outcomes |
| ☒ | ☐ | Estimates of effect sizes (e.g. Cohen's *d*, Pearson's *r*), indicating how they were calculated |

*Our web collection on statistics for biologists contains articles on many of the points above.*

## Software and code

Policy information about availability of computer code

| Data collection | OC-725C oocyte clamp amplifier (Warner Instruments)<br>Titan Krios (Thermo Fisher Scientific), K3 Summit camera (Gatan)<br>luminometer (Centro, LB960) |
|---|---|
| Data analysis | EPU 2 (Thermo Fisher Scientific) 2.8.1.10REL<br>Relion 3.1<br>Coot 0.9<br>PHENIX 1.18.2<br>USCF Chimera 1.15<br>ChimeraX 1.15<br>OriginPro 2022<br>pClamp 10.6<br>Pymol Molecular Graphics System 1.7.2.1.<br>GraphPad Prism 8<br>Microsoft Office Software package (Excel) 2016<br>RStudio 2021.09.0 Build 351 |

For manuscripts utilizing custom algorithms or software that are central to the research but not yet described in published literature, software must be made available to editors and reviewers. We strongly encourage code deposition in a community repository (e.g. GitHub). See the Nature Portfolio guidelines for submitting code & software for further information.

## Data

Policy information about availability of data

All manuscripts must include a data availability statement. This statement should provide the following information, where applicable:

- Accession codes, unique identifiers, or web links for publicly available datasets
- A description of any restrictions on data availability
- For clinical datasets or third party data, please ensure that the statement adheres to our policy

The atomic coordinates of the Sr35 resistosome have been deposited in the Protein Data Bank with the accession code 7XC2. The EM map for the local mask of Sr35 LRR in complex with AvrSr35 has been deposited in the EMDB with the accession code EMD-33111.
Sequences of TaSh1 and HvSh1 are available at NCBI under accession codes XP_044359492.1 (https://www.ncbi.nlm.nih.gov/protein/XP_044359492.1/) and KAE8803279.1 (https://www.ncbi.nlm.nih.gov/protein/KAE8803279.1 ), respectively.
Source data of tobacco infiltrations, western blots, insect cell viability and wheat protoplast cell death are provided with this manuscript.

# Field-specific reporting

Please select the one below that is the best fit for your research. If you are not sure, read the appropriate sections before making your selection.

☒ Life sciences   ☐ Behavioural & social sciences   ☐ Ecological, evolutionary & environmental sciences

For a reference copy of the document with all sections, see nature.com/documents/nr-reporting-summary-flat.pdf

# Life sciences study design

All studies must disclose on these points even when the disclosure is negative.

| | |
|---|---|
| Sample size | No statistical methods were used to determine sample size. Sample size was chosen in accordance with the generally accepted standard of the respective scientific field.<br>The wheat protoplast experiments consisted of a total of five to six replicates. A maximum of two replicates were conducted per batch of wheat seedlings to encompass any variation due to the plant material. Five to six replicates of the wheat protoplast experiments were deemed sufficient due to the consistency of the results. The N. benthamiana infiltration experiments consisted of a minimum of seven biological replicates. Seven replicates are deemed sufficient due to any variation of protein expression in the leaves.<br>Sample size of Xenopus experiments was chosen based on previous literature (Bi et al. 2021) and deemed sufficient due to the low variation between technical and biological replicates.<br>More information is given in Statistics and reproducibility. |
| Data exclusions | No data were excluded from the analyses. |
| Replication | Generally, repetition was a measure taken to combat experimental variation. The plant data is deemed reproducible due to the use of different batches of plant material in both the wheat protoplast experiments and the N. benthamiana experiments. In addition, different Agrobacterium cultures were used to verify reproducibility of the N. benthamiana experiments.<br>The Xenopus data is deemed reproducible due to the use of different oocyte batches.<br>Expression from Sf21 insect cells was generally reproducible whenever the cell culture was in good health. Recovery of star-shaped particles varied according to culture health, baculovirus quality and experimenter performance in protein purification. Generally >20% star-shaped particles were recovered from Sr35 L15E/L19E and AvrSr35 co-expression.<br>Negative staining and cryo grid preparation are tricky procedures. Variablity of cryo grid preparation was reduced by using Vitrobot automation. Nevertheless, recovery of high quality electron microscopy samples varied significantly. |
| Randomization | Plant material was selected randomly from a given batch and analyzed equally. Randomization was deemed unnecessary as no sub-sampling was done. |
| Blinding | Blinding was not deemed relevant for our experiments given the nature of the reagents (plasmids, cRNA) or due to the protein purification work-flow (generally one large-scale purification at a time). |

# Reporting for specific materials, systems and methods

We require information from authors about some types of materials, experimental systems and methods used in many studies. Here, indicate whether each material, system or method listed is relevant to your study. If you are not sure if a list item applies to your research, read the appropriate section before selecting a response.

## Materials & experimental systems

| n/a | Involved in the study |
|---|---|
| ☐ | ☒ Antibodies |
| ☐ | ☒ Eukaryotic cell lines |
| ☒ | ☐ Palaeontology and archaeology |
| ☐ | ☒ Animals and other organisms |
| ☒ | ☐ Human research participants |
| ☒ | ☐ Clinical data |
| ☒ | ☐ Dual use research of concern |

## Methods

| n/a | Involved in the study |
|---|---|
| ☒ | ☐ ChIP-seq |
| ☒ | ☐ Flow cytometry |
| ☒ | ☐ MRI-based neuroimaging |

# Antibodies

| | |
|---|---|
| Antibodies used | monoclonal mouse Anti-myc ( R950-25, Invitrogen/Thermofisher)<br> c-Myc synthetic peptide: Glu-Gln-Lys-Leu-Ile-Ser-Glu-Glu-Asp-Leu-<br> Species reactivity: Tag.  no further validation data available from manufacturer<br><br>polyclonal goat anti-mouse IgG-HRP (ab6728, Abcam), no further validation data available from manufacturer<br><br> polyclonal rabbit Anti-GFP; ( pabg1, Chromtek), no further validation data available from manufacturer<br><br>polyclonal swine anti-rabbit IgG-HRP (PO399, Agilent DAKO) |
| Validation | Antibodies were not validated in-house and not validated by the manufacturer (see above).<br><br>However, antibodies were deemed reliable due to entries in 'antibodyregistry' under entries: monoclonal mouse Anti-myc (RRID = AB_2556560), polyclonal goat anti-mouse IgG-HRP (RRID =  AB_955440) and polyclonal rabbit Anti-GFP (RRID =  AB_2749857).<br><br>In addition, antibodies were validated by the use of an empty vector negative control (to control unspecific binding of antibodies to tobacco/agrobacterium proteins) and wild-type Sr35 (myc-tagged) or wild-type AvrSr35 (YFP-tagged) as positive controls on each western blot. |

# Eukaryotic cell lines

Policy information about cell lines

| | |
|---|---|
| Cell line source(s) | Sf21 insect cell line (Invitrogen) |
| Authentication | none of the cell lines were authenticated |
| Mycoplasma contamination | cell line was not tested for mycoplasma contamination |
| Commonly misidentified lines (See ICLAC register) | no commonly misidentified cell lines were used in this study |

# Animals and other organisms

Policy information about studies involving animals; ARRIVE guidelines recommended for reporting animal research

| | |
|---|---|
| Laboratory animals | Xenopus laevis, female, age 2-2.5 years |
| Wild animals | no wild animals were used in this study |
| Field-collected samples | no field collected samples were used in this study |
| Ethics oversight | The animal study (Xenopus laevis) was reviewed and approved by Laboratory Animal Ethics Committee at Institute of Genetics and Developmental Biology, Chinese Academy of Sciences, Beijing, China with the approval ID AP2020029. |

Note that full information on the approval of the study protocol must also be provided in the manuscript.

