## [Peer Review File · Nature]

Manuscript Title: A wheat resistosome defines common principles of immune receptor channels

Reviewer Comments & Author Rebuttals

Reviewer Reports on the Initial Version:

Referees' comments:

Referee #1 (Remarks to the Author):

Plant intracellular NLRs recognize pathogen effectors to trigger plant ETI. The group solved the first structure of the plant NLR complex, termed ZAR1 resistosome, which functions as a calcium channel required for ZAR1-mediated ETI. In this manuscript, the authors reported the cryo-EM structure of wheat NLR Sr35 in complex with the effector AvrSr35. The structure of wheat Sr35 and Arabidopsis ZAR1 resistosomes show a striking similarity. Like ZAR1 resistosome, AvrSr35-Sr35 complex exhibits non-selective cation channel activities. Both Sr35 and ZAR1 are CNLs. The studies reveal the evolutionary conservation of CNL resistosomes in activating plant immunity. Furthermore, this study reports an arginine cluster in the LRR domain that co-evolves and forms intramolecular interaction with CC EDVID motif. C-terminal LRR of Sr35 recognizes AVRsr35. Interestingly, structure-based domain swapping and mutational analysis of orphan CNLs bearing homology with Sr35 generate gain-of-function of CNLs that recognize AvrSr35, suggesting the potential of structure-based engineering of NLRs for crop improvement. The studies not only reveal the conservation of plant CNL resistosomes, but also provide novel insights into plant NLR activation and potential application in crop improvement. The manuscript provides a conceptual advance in understanding plant NLR activation. It was well written with high-quality data.

Here are some suggestions for authors to improve the manuscript.

1. Line 56, what is the meaning of “the latter receptors”? It is also not clear the meaning of “share strain-specific immunity with Sr genes”.
2. Line 57-60. Grammar mistake. Delete “was” in line 57. What is the genome of “Triticum monococcum”? “it’s a-genome diploid donor T. Urartu”. “It” means bread wheat or Triticum monococcum.
3. Line 66. It seems that Fig. 1a does not have cell death data.
4. Line 121-122, and Figure 2e, its CC L141 and Y42 in the figures, but Y141 and L42 in the text. In Fig. 2g-j, they were labeled Y141 and L42.
5. The authors found the co-occurrence of the EDVID motif and LRRR-cluster that mediate intramolecular interactions to stabilize CNL resistosomes. They also show the functional importance of this motif in triggering cell death. It will be more convincing to use Co-IP or gel filtration to show

the involvement of these motifs in intramolecular interactions.

6. Fig 3 a, b. does coexpression of Sr35 and AvrSr35 induce cell death in oocytes? If not, why? They induce cell death in insect cells. 3b, explain ND96 in the text or legend. How about LaCl₃ treatment itself in 3a and 3b? The claim that these results suggest that Sr35 resistosome has calcium channel activity in *Xenopus* oocytes in line 205 is not strongly backed by the data. Suggest deleting or toning down here.

7. It is interesting, also puzzling, that the acidic residues in N-terminus are not required for Sr35 channel activity and functions. The Sr35R730D/R755Q and Sr35W803L/K754G mutants blocked channel activity. But these residues are in the C-terminus and may not be directly involved in channel formation based on the ZAR1 model. The authors proposed that the very N-terminus of the Sr35 resistosome (residues 1 to 21) is structurally and functionally distinct from that of the ZAR1 resistosome. Notably, both AvrSr35 and Sr35 are required to trigger channel activities. Did the authors test whether CC domain of Sr35 itself is sufficient to trigger channel activity? If yes, whether acidic residues are required?

Referee #2 (Remarks to the Author):

Forderer et al report the activated form of the plant NLR Sr35 from wheat in complex with the effector AvrSr35 (the resistosome). This is only the second structure of a resistosome of a plant NLR from the CC class (the first one being the resistosome of the Arabidopsis NLR ZAR1). In this way this is a very exciting work that in particular tests the generality of some implications of the ZAR1 structure. Some key new observations brought by this new work involve:

- The first structure of a CC-NLR that recognizes its effector by direct binding (ZAR1 displays indirect recognition)
- The first structure of a CC-NLR from monocots shows that distantly related CC-NLRs from monocots and dicots form pentameric resistosomes
- The similarity extends to the channel structure formed in the resistosome by the CC domain and the corresponding cation channel function.
- Engineering of related barley receptors allowed generation of variants that recognize AvrSr35.
- AvrSr35 structure exhibits a new helical fold.

Having said all this, the paper leaves a feeling that the authors could have made this a more exciting paper, and there are a few disappointing aspects.

- The data on the calcium channel activity and selectivity is not conclusive, and the authors leave in rather unexplained (the text "...the channel activity of the Sr35 resistosome is tolerant to substitutions of acidic residues predicted to line the inner surface of the channel. Thus, we cannot exclude the possibility that the very N-terminus of the Sr35 resistosome (residues 1 to 21) is structurally and functionally distinct from that of the ZAR1 resistosome" more sounds like trying to avoid explaining the data). Why was the L15E/L17E mutant not investigated in the oocyte system?
- It is not clear if the approach used in section "Steric clash..." is the best approach; why use

AlphaFold2 prediction, when one has no control over what conformation of a protein that exists in more than one conformation, such as a plant NLR, is used to derive the predicted model? It would make much more sense to use the inactive ZAR1 structure as a template and derive the model by superimposing the individual Sr35 domains and derive a physically plausible structure by energy minimizing such a model, one would then have much better control over the model in that way. I suspect the conclusions will be the same, but the approach used here appears somewhat hit-and-miss.

- The abstract mentions a “previously unnoticed arginine cluster” interacting with the EDVID motif; this is incorrect - the structural basis of function of the EDVID motif has been discussed previously in PMID: 31415752, which is not cited in this manuscript

- The cryo-EM density of the reconstruction is of high quality, but the corresponding structure model appears unfinished and can easily be improved (see details below). When sidechains are not visible, modelling them as alanines or glycines will just lead to confusion by researchers using the model; it would be much better to model the most likely rotamer (or alternatively keep the correct sequence but omit parts of the sidechain that are not visible, although this will undoubtedly also lead to confusion by researchers using the model).

Nevertheless, this is important work and hopefully the authors can improve the paper and the structure models in the final version of the paper.

Problems with the structural model and structural analysis

- Section “Sr35 oligomerization and stabilization features...” paragraph 2: hydrophobic contacts of L47 with Y141 are discussed. This appears to be incorrect, as L47 is not in contact with Y141. The authors may be referring to V46?

- Structure model: AvrSr35

- o Residue 130: Appears to be modeled incorrectly.

- o Residues 169-170: Missing residues. There is clear map/volume for the main chain in this part and this small chain break could be modeled.

- o Residues 171-174: Modeled as polyalanine, not AvrSr35 sequence. This should be modeled as the correct sequence. If uncertain of sidechain positions, either remove sidechains or select most appropriate rotamer.

- o Residues 202-206: Modeled as glycine/alanine, not AvrSr35 sequence.

- o Residues 206-209: Missing residues can likely be modeled. The map in this region is still clear and distinct at lower threshold.

- o Residues 200-209: Poorly modeled.

- o Residues 244-250: Missing residues can likely be modeled. The map in this region is clear and distinct at lower threshold.

- o Residues 273-277: Model slightly out of map register. Fit could be improved.

- o Residues 376-378: Sidechains appear modeled incorrectly.

- o Residues 486-489: Incorrectly modeled.

- Structure model: Sr35 molecule

- o Residues 111-112: Incorrectly modeled. Residues do not fit map, even at low threshold.

- o Residues 346-353: Appears incorrectly modeled. Missing residues 348-352 could be built. There is clear map at lower threshold for the main chain of these missing residues.

- o Residues 564-571: Multiple issues in this region of the structure. Polyalanine is modeled instead of the correct sequence, despite many similar residues (valine and glycine) in the actual sequence.

Residues 568-570 are missing, despite the structure appearing as a continuous chain with residue 567 forming a peptide bond with residue 571.

o Residue 862: Incorrect sequence.

o Residues 864-868: Sequence is incorrect. Residues modeled as alanine.

- There are many rotamer outliers that are either not supported by the map or have alternative non-outlier positions that are equally well fitting. These outliers should be addressed. Some examples:

o AvrSr35: 201, 275, 283, 421, 425, 534.

o Sr35: 161, 178, 188, 199, 250, 315, 335, 359, 582, 624, 639, 671, 678, 682, 707, 786.

Minor comments on the manuscript

1. Main paragraph 1 line 7: "NODs"

2. Section "Sr35 oligomerization and stabilization features..." page 2 paragraph 3 line 4: delete ")"

3. Section "Sr35 oligomerization and stabilization features..." page 2 paragraph 3 line 10: "LRR" stands for "leucine-rich repeat", so "LRR repeats" is redundant.

4. Section "Sr35 oligomerization and stabilization features..." page 2 paragraph 3 line 11 and elsewhere: there is no such thing as "primary (amino acid) sequence" – there is no secondary sequence is there? The correct term is "primary structure".

5. Section "The Sr35 resistosome has nonselective..." paragraph 1 line 9: "AvrSr35" is misspelled.

6. Section "The Sr35 resistosome has nonselective..." page 2 paragraph 2 line 6 and throughout: "tetrabutylammonium" probably does not have to be capitalized. Check capitalization throughout, "Electrophysiology" section in particular.

7. Materials and Methods section "Cryo-EM sample..." line 11 and elsewhere: put space between number and unit ("50 e/A²").

8. Materials and Methods section "Image processing..." line 4: delete ")".

9. Materials and Methods section "Image processing...": why were the two datasets not combined?

10. Materials and Methods section "Transient..." page 2 line 1: "sequences"

11. Materials and Methods section "Statistical..." line 10: delete underscore.

12. References: need to be checked and format made consistent: many are missing the journal (e.g. 6-10 etc), article title and journal title capitalization is inconsistent, journal titles are sometimes abbreviated and sometimes not.

13. Figure 2 legend line 3: "indicates"

14. Figure 2 legend line 16 and elsewhere: "methods" should be "Materials and Methods".

15. Figure 4 legend line 1: "LRR domain"

16. Figure 4 legend line 5-6: "Relative...death" is duplicated.

17. Figure 5 legend page 2 line 5: "chimera"

18. Extended Data Fig. 1 legend: there is no "c" ("b" is used twice") and neither says what is actually shown: chromatogram, gel of some sort?

19. Extended Data Fig. 5 legend line 5: "Detailed"

20. Extended Data Fig. 11 legend line 1: "ROQ1 and RPP1 ligand binding sites". What is "ascending lateral LRR domain"?

21. Extended Data Fig. 12: say which structures are predicted and how were they derived.

22. Extended Data Table 1: are these "Selected data from the PDB validation report", but they are not just validation data, they are structure determination parameters? The lines are misaligned in much of the table.

Referee #3 (Remarks to the Author):

This manuscript presents a detailed structural analysis of a plant immune receptor-pathogen ligand complex that makes a substantial advance in our understanding of how plants resist disease. The authors find that when bound to its ligand avrSr35, the Sr35 receptor forms a pentameric complex and exhibits cation channel activity. This structure and cation channel activity is similar to that of ZAR1 previously reported by this group, which is the only other structure for an immune receptor of this class (CC-NLR) and derives from a distantly related species. Thus, this work provides evidence that these common features are in fact most likely general characteristics of all or most plant receptors of this class rather than idiosyncrasies of an individual receptor, therefore suggesting common functionalities. Extensive mutational analyses confirm the role of specific residues in effector recognition and oligomerisation as suggested by the structure. Modelling of the inactive state of the receptor further suggests that effector binding to the LRR domain favors the active state through a steric clash with the central NB domain. Swapping of the Sr35 LRR onto distantly related immune receptors allows their activation by AvrSr35, suggesting that this model of receptor activation may be common. Overall, this is a major advance in our understanding of plant immunity, and also adds to the growing understanding of similar classes of immune receptors in humans and other animals and this manuscript can be expected to have substantial and lasting impact in the field.

The methodology applied here is robust, appropriate statistical analyses have been included and the conclusions drawn are based on solid data and reasoning and appear robust. The article is well written and cites prior literature appropriately. I have no suggestions for additional data or experiments, but have made some comments below on some points of interpretation, reasoning, clarification or presentation that the authors should consider.

Some points for the authors to consider.

Lines 118 to 125. In the ZAR1 structures there was a difference between the CC domains in the inactive and active states in the alpha helix 4 region – designated 4a and 4b in the inactive and active structures. CC-alone structures are similar to the inactive ZAR1 state. Could the authors discuss which of these structures is more similar to the Sr35 CC here in terms of the alpha 4 helix? It is hard to tell from the figure, but it looks like there is only a short loop between helices 3 and 4 in this case, which is more like the inactive ZAR1 structure, and those of CC-alone structures. Do the authors think that this change in helix 4 during activation is a common feature of both ZAR1 and Sr35 or perhaps is specific to ZAR1?

Line 182-183. Is it necessarily the case that this interaction ‘must’ be disrupted to allow the a1 helix to flip out? Does this assume that the helix itself remains intact during the process? Could not the a1 helix unfold and then refold after moving out? Especially given that this part of the Sr35 structure is not resolved.

Lines 186. This could be rephrased. Rather than the funnel-shaped structure being ‘not well defined’,

it is the N-terminal region of Sr35 that is not well-defined while the funnel structure is absent. Eg. "...although the funnel-shaped structure composed of the α 1-helix in the ZAR1 resistosome was not seen in the Sr35 resistosome structure, the corresponding region of Sr35 was not well-defined and apart from this the two structures are highly similar. We thus reasoned..." Could the authors also comment on why this structure is absent – is it likely that this is because of the L15E/L19E double mutation in this region which was used to circumvent cell death in insect cells? Presumably that works by preventing proper function of the α 1 helix, without affecting structure of the rest of the protein protein or its ability to recognise the effector and transform to an active state as observed here. It seems likely that those mutations could primarily affect funnel formation.

Lines 202-204. Some further explanation of the reasoning here would be useful. Is the A01 inhibitor expected to completely block all endogenous xenopus Ca channel activity? Hence the remaining activity is due to the Sr35 channel not being affected by this inhibitor due to its non-canonical nature? LaCl₃ is a non-specific inhibitor as a Ca mimic?

Line 256-260. This section seems to be presented in a manner that proceeds from an assumption, rather than analysing a result and making a conclusion. What is the justification for claiming this is a model of inactive Sr35? What confidence do we have that it is correct? Is it similar to the inactive state of ZAR1? I think this should be presented in a different logical order along the lines of the following – the authors could express this more precisely I'm sure: "We used alphafold2 to predict a structure for Sr35 in the absence of AvrSr35 and compared this to the resistosome structure. Many parts were similar (eg LRR), suggesting that alphafold did a good job of local structure prediction, but the NOD module showed substantial organisational differences. (((The modelled structure was similar to the ZAR1 inactive state (???), suggesting that it is likely to be a good representation of the Sr35 inactive state – note that if it is not similar to inactive ZAR1, then we'd be concerned about whether it is correct.)) Modelling of AvrSr35 onto the LRR domain of the predicted Sr35 structure shows substantial overlap between the effector and the Sr35 NBD. Assuming the modelled structure represents the inactive state of Sr35, this steric clash..."

Line 267-268. Replace 'appears a conserved mechanism of plant NLRs' with " is a common feature of the four known resistosome structures (ED fig 11), suggesting it may be a conserved mechanism of plant NLRs." Given this conclusion, could the authors elaborate a little on why this should be so? It seems to me that the 'lateral ascending side' of the LRR is the one that is adjacent to the NOD region in the known structures. Thus, effector binding here is more likely to lead to a steric clash with the inactive conformation. Is this a reasonable argument to propose? Binding on the other side could potentially occur irrespective of the activation state, hence would not predispose the receptor towards either state.

Line 317 to 320. The reasoning presented here could be refined a little. Firstly, it should be suggested that 'the activation and signalling mechanisms of CNLs are evolutionarily conserved' (we know the CNLs themselves are evolutionarily related). Of the 'Two lines of evidence' discussed, one refers to exchanges between wheat and barley NLRs. Since these are both monocots it does not bear on the conservation between monocots and dicots. Conversely, the subsequent statement that 'this conservation extends to cation fluxes' constitutes an additional (third) line of evidence. Perhaps just delete 'monocotylenous and dicotyledonous' and then cite three lines of evidence. The key part of

the wheat Sr35 and barley MLA swap is not so much that it is between species, but that the genes are not orthologous. Swapping between wheat and barley orthologs may not be much different than between alleles, which has been done extensively previously, but this represents a further step as the NLRs are less related.

Minor edits.

Line 55. Reference 15 Yu et al 2016 describes genetic mapping of Sr46 not the gene cloning. Should cite here Arora et al. 2019, Resistance gene cloning from a wild crop relative by sequence capture and association genetics. *Nature biotechnology*, 37, 139–143. <https://doi.org/10.1038/s41587-018-0007-9>

Line 66 and line 73. Reference to Fig1a should be to extended data fig 1a.

Line 69. The Sr35-L15E/L19E mutant is not named in ED fig 1a. Is this the sample labelled deltaOS in the figure?

Line 78. This should be rephrased as the UV absorbance is not 550kDa (70 mL), but the peak of UV absorbance occurs at an elution volume of 70mL corresponding to a size of 550kDa. However, please check this versus the figure which seems to show the peak at about 65 mL and labels this with an arrow indicating a size of 900kDa.

Extended data figure 1c. Not all of the lanes are labelled. Mostly it would seem that the missing ones are continuous ie 63 between 62 and 64, but this is not so for all the lanes at the right.

Line 80 reference to Figure 1a should also include fig 1b

Line 84 reference to figure 1b here does not seem relevant.

Line 139 and Fig 2g. The text states that an L42E mutation was made, but the protoplast section of the figure is labelled L42A, L42E is in the benthamiana part of the figure. ED Fig4a also has L42A

Line 164. Sr35 should be ZAR1.

Line 150-170 – note that the Rairdan et al paper that defined the EDVID motif (ref 24) also actually showed that it had a role in mediating interaction between the CC domain and the NB-LRR, which is consistent with the structural findings here.

Line 174. Fig 2h,i should refer to fig 2i,j.

Line 177 delete respectively.

Line 221. The term ‘ascending lateral side’ may not be clear to non-LRR specialists.

Line 251. This is a little vague, but it would be worth to point out more specifically that most of the single amino acid mutations had no effect on AvrSr35 recognition, while the single mutations Y387A

and R395A did have a partial effect on reducing recognition but did not abrogate it entirely. Also the Q72* mutant (a premature stop codon?) is not mentioned.

Line 283, 296. These are not 'single' substitutions. There were either 8 or 10 substitutions made in each case. Perhaps rephrase as "targetted mutations of specific residues".

Fig2 J "Y74A/E77A/D78A" – the A is missing after Y74 in this label.

Figure 4 legend, repeated sentence "Relative luminescence was used as a readout for cell death"

ED Fig1b,c figure legend is not informative eg fig1b shows a size exclusion chromatography profile. And 1c (labelled b again in legend) shows a stained electrophoresis gel.

Referee #4 (Remarks to the Author):

In this study, Förderer, Li, Lawson, Deng and colleagues developed a high-resolution structure of the wheat NLR immune receptor protein Sr35 in complex with its corresponding effector AvrSr35. They find that the active immune receptor complex forms a higher order pentameric complex which has structural features similar to that of the recently resolved ZAR1 complex from the model plant *Arabidopsis thaliana*. The authors performed frog oocyte electrophysiological assays to show that the complex acts as a non-selective cation channel. In a series of elegant experiments, the authors then used the structure to guide the engineering of Sr35 homologues from wheat and barley which do not recognize AvrSr35. They demonstrate that a small number of amino acid substitutions in Sr35 (which are surface exposed and make contact with AvrSr35) are enough to convert these wheat and barley homologues into gain-of-function variants that mount an effective AvrSr35-mediated immune response.

This is a tour-de-force study encompassing cutting edge cryo-electron microscopy. We learn that coiled-coil NLRs separated by 200 million years of evolution (i.e. defining the major split in land plants between monocots and dicots) form similar oligomeric immune complexes. The deft structure-guided approach to engineering new-to-nature functional immune receptors has potential important applications in wheat disease resistance breeding as the few amino acid changes required (8 to 10) would allow these variants to be introduced into the wheat genome by gene editing and subsequently deployed in several jurisdictions including the USA, Canada, Australia, China and South America, where these type of biotech crops are now accepted.

The manuscript is exceptionally well written with highly didactic figures. I expect the study will gain the strong attention of the communities studying molecular plant-microbe interactions, cereal disease resistance, wheat breeding, crop protection biology, genetic engineering, and structural biology. In conclusion, I am highly supportive of this study. In making suggestions for improvements, I have only a few minor comments, which can all be addressed without the need for additional experimentation, as noted below:

1. The study has generated a large number of constructs. Please consider making these available in Addgene as this will be a great resource for the scientific community.

2. Page 2: The authors state that: “Wheat stem rust caused by the fungal infection with *Puccinia graminis* f. sp. *tritici* (Pgt) threatens global food production [...] has motivated the search for stem rust resistance genes [...]. This resulted in the isolation of eight stem rust resistance (Sr) genes that are phylogenetically related and belong to a clade of grass CNLs all of which confer strain-specific immunity⁹⁻¹⁵.”

In response to this statement please note the following:

(i) The citation should also include reference number 16 which describes the cloning of Sr26 and Sr61 which encode CNLs. If I am not mistaken, that brings the total tally of to ten i.e. Sr13, Sr21, Sr22, Sr26, Sr33, Sr35, Sr45, Sr46, Sr50 and Sr61.

(ii) You might also consider including SrTA1662, cloned from *Aegilops tauschii*, and introgressed into wheat, see Gaurav et al., 2022: <https://www.nature.com/articles/s41587-021-01058-4>
This would bring the total tally to 11.

(iii) The reference for Sr46 cloning (Yu et al., 2015) is wrong. This paper provides genetic characterization of Sr46. The cloning paper is Arora et al., 2019: <https://pubmed.ncbi.nlm.nih.gov/30718880/>

(iv) The way the sentence is currently phrased one is led to believe that all the cloned Sr genes are CNLs. However, the same motivation that led to the cloning of the CNL Sr genes also led to the cloning of Sr60 and Sr62, which in fact encode wheat-tandem kinases, see: <https://pubmed.ncbi.nlm.nih.gov/31487050/>
<https://pubmed.ncbi.nlm.nih.gov/35338132/>

(v) Finally, please note that “f sp.” should be “f. sp.”

3. On page 3, the authors introduce the experiment in which Sr35 and AvrSr35 are expressed in insect cells. They go on to state: “Unexpectedly, cell death was observed when the receptor was co-expressed with AvrSr35 (Fig. 1a)...” A reference to this result is also made in the Discussion on page 10. However, I could not see these insect cell death results depicted in Figure 1a. Perhaps the authors meant to cite Supplementary Figure 1a? In this figure, please define explicitly the y-axis.

4. On page 4, “known proteins structures” should be “known protein structures”.

5. Top of page 7. Please consider spelling out the abbreviation/acronym of “CaCC” – I think this is the first use of the abbreviation.

6. Page 8. “with Sr35 LRR” should be “with the Sr35 LRR”.

7. Figure 3, page 24. The figure legend starts with the abbreviation TEVC – please consider spelling

this out in full.

8. Figure 4. Why is the R395A mutation (tested in *N. benthamiana* in panel f) not highlighted in panel b?

Author Rebuttals to Initial Comments:

We would like to take a moment and thank all reviewers for the constructive criticism. We greatly appreciate the diverse panel of expertise that they brought to the peer review process and are pleased with the improvements they have allowed us to make on the manuscript. Below we have prepared a point-by-point reply.

Referees' comments:

Referee #1 (Remarks to the Author):

Plant intracellular NLRs recognize pathogen effectors to trigger plant ETI. The group solved the first structure of the plant NLR complex, termed ZAR1 resistosome, which functions as a calcium channel required for ZAR1-mediated ETI. In this manuscript, the authors reported the cryo-EM structure of wheat NLR Sr35 in complex with the effector AvrSr35. The structure of wheat Sr35 and Arabidopsis ZAR1 resistosomes show a striking similarity. Like ZAR1 resistosome, AvrSr35-Sr35 complex exhibits non-selective cation channel activities. Both Sr35 and ZAR1 are CNLs. The studies reveal the evolutionary conservation of CNL resistosomes in activating plant immunity. Furthermore, this study reports an arginine cluster in the LRR domain that co-evolves and forms intramolecular interaction with CC EDVID motif. C-terminal LRR of Sr35 recognizes AVRsr35. Interestingly, structure-based domain swapping and mutational analysis of orphan CNLs bearing homology with Sr35 generate gain-of-function of CNLs that recognize AvrSr35, suggesting the potential of structure-based engineering of NLRs for crop improvement. The studies not only reveal the conservation of plant CNL resistosomes, but also provide novel insights into plant NLR activation and potential application in crop improvement. The manuscript provides a conceptual advance in understanding plant NLR activation. It was well written with high-quality data.

Here are some suggestions for authors to improve the manuscript.

1. Line 56, what is the meaning of “the latter receptors”? It is also not clear the meaning of “share strain-specific immunity with Sr genes”.

We rewrote the sentence and specified what we mean with ‘the latter’. We believe that in the new context it is now clearer that ‘share strain-specific immunity with Sr genes’ refers to *Mla* and *Sr* genes.

2. Line 57-60. Grammar mistake. Delete “was” in line 57. What is the genome of “Triticum monococcum”? “it’s a-genome diploid donor T. Urartu”. “It” means bread wheat or Triticum monococcum.

We rewrote the section to better explain the relationship between *T. urartu* and *T. monococcum*.

3. Line 66. It seems that Fig. 1a does not have cell death data.

We meant to write Extended Data Fig 1a. We made the corresponding change in the text.

4. Line 121-122, and Figure 2e, its CC L141 and Y42 in the figures, but Y141 and L42 in the text. In Fig. 2g-j, they were labeled Y141 and L42.

It should be Y141 and L42 (with Y141A and L42E being the corresponding point mutations). The errors have been corrected.

5. The authors found the co-occurrence of the EDVID motif and LRRR-cluster that mediate intramolecular interactions to stabilize CNL resistosomes. They also show the functional importance of this motif in triggering cell death. It will be more convincing to use Co-IP or gel filtration to show the involvement of these motifs in intramolecular interactions.

We agree that more insight into the effect of R-cluster/EDVID mutants will be useful. To clarify the point raised, we performed several experiments. (1) Blue native (BN) PAGE of *N. benthamiana* leaf extracts and (2) size exclusion chromatography of Sr35 and AvrSr35 proteins expressed in Sf21 insect cells.

In reference to BN PAGE (1), we tried the standard protocol used by other authors (e.g. Ahn et al. 2022, Ayutthaya et al. 2020). Using the standard buffer (pH 6.8) for preparing a lysate from *N. benthamiana* leaves for BN PAGE, the amount of solubilized Sr35 protein was insufficient for detection. Next, we tried the lysis buffer that we had previously used to solubilize Sr35 from insect cells (pH 8). SDS PAGE of this lysate showed that significantly more Sr35 can be solubilized with this buffer, but we still failed to detect Sr35 oligomers on BN PAGE. We suspect that the low pH of the BN system (pH 6.8) interferes with the solubility and/or stability of Sr35 oligomers.

We then tried expression of EDVID/R-cluster mutants in Sf21 insect cells (2) and are pleased to report the following new data (Figure 1).

Figure 1 Purification of EDVID/R-cluster mutants co-expressed with AvrSr35 from Sf21 insect cells. *estimated oligomer peak, **estimated heterodimer peak

We introduced L15E/L19E substitutions to the EDVID and R-cluster mutants to avoid cell death in insect cells and co-expressed these receptor constructs with AvrSr35. We used GST-tag (AvrSr35) to bind Sr35 receptor/receptor variants to glutathione beads and cleaved the protein from the beads using PreScission protease. Gel filtration analysis of these protein complexes revealed that both the EDVID and R-cluster mutants retained their activity for interaction with AvrSr35. This provides further evidence that the C-terminal LRR domain is exclusively responsible for AvrSr35 recognition. As anticipated, all Sr35 L15E/L19E protein formed a complex with AvrSr35 (the Sr35 resistosome) and eluted at a position with a molecular weight of 669 kDa (fractions ~56-72 mL), indicating L15E/L19E have no impact on the assembly of the Sr35 resistosome. By contrast, introduction of mutations in the EDVID motif or the R-cluster shifted a small fraction of Sr35 protein to lower molecular species (~76-84 mL), supporting a role of the EDVID motif and the R-cluster in the assembly of the Sr35 resistosome. However, the same mutations in the EDVID motif or the R-cluster significantly compromised cell death in tobacco and wheat protoplasts. How can these biochemical and genetic data be reconciled? As discussed in the manuscript, oligomerization of the Sr35 resistosome is mainly mediated by the central NOD module. The intramolecular interactions between the EDVID motif and the R-cluster act to stabilize the conformation of the CC domain to form a helical barrel, which further contributes to the stabilization of the Sr35 resistosome. Mutations at the EDVID motif or the R-cluster are predicted to destabilize the conformation of the CC domain and consequently disrupt the helical barrel where the potential calcium channel of the Sr35 resistosome is located. By contrast, these mutations will not impact NOD-mediated oligomerization of the Sr35 resistosome, explaining why the mutant proteins were only slightly compromised in oligomerization. For simplicity, we decided not to include these new data in the revision. However, we are happy to include them in the manuscript at the request of the reviewers.

Ahn, H. K., Lin, X., Olave-Achury, A. C., Derevnina, L., Contreras, M. P., Kourelis, J., ... & Jones, J. D. (2022). Effector-dependent activation and oligomerization of NRC helper NLRs by Rpi-amr3 and Rpi-amr1. *bioRxiv*.

Na Ayutthaya, P. P., Lundberg, D., Weigel, D., & Li, L. (2020). Blue native polyacrylamide gel electrophoresis (BN-PAGE) for the analysis of protein oligomers in plants. *Current Protocols in Plant Biology*, 5(2), e20107.

6. Fig 3 a, b. does coexpression of Sr35 and AvrSr35 induce cell death in oocytes? If not, why? They induce cell death in insect cells. 3b, explain ND96 in the text or legend. How about LaCl₃ treatment itself in 3a and 3b? The claim that these results suggest that Sr35 resistosome has calcium channel activity in *Xenopus* oocytes in line 205 is not strongly backed by the data. Suggest deleting or toning down here.

Indeed, we observed that Sr35 and AvrSr35 co-expression eventually leads to cell death in *Xenopus* oocytes after completion of our electrophysiological measurements, but we never quantified this effect. We conducted the current measurement 4~8 h after oocyte injection, whereas apparent cell death occurs only after overnight incubation (>16 hours).

We added information on the ND96 solution to figure legend 3b. ND96 solution is a complete culture solution containing 96 mM NaCl, 2.5 mM KCl, 1 mM MgCl₂, 1.8 mM CaCl₂, 5 mM HEPES pH 7.6.

We have now added only LaCl₃ treated samples in Figure 3 a and b.

We changed the sentence in line 206 and hopefully found a more cautious phrasing: 'Together, these results suggest that the Sr35 resistosome may contribute to mixed currents in *Xenopus* oocytes, possibly via Sr35 calcium channel activity.'

7. It is interesting, also puzzling, that the acidic residues in N-terminus are not required for Sr35 channel activity and functions. The Sr35R730D/R755Q and Sr35W803L/K754G mutants blocked channel activity. But these residues are in the C-terminus and may not be directly involved in channel formation based on the ZAR1 model. The authors proposed that the very N-terminus of the Sr35 resistosome (residues 1 to 21) is structurally and functionally distinct from that of the ZAR1 resistosome. Notably, both AvrSr35 and Sr35 are required to trigger channel activities. Did the authors test whether CC domain of Sr35 itself is sufficient to trigger channel activity? If yes, whether acidic residues are required?

As discussed in the manuscript, the mechanism underlying the dispensability of the acidic residues in the N-terminus remains unknown. The mutations of R730D/R755Q and W803L/K754G in the C-terminus of Sr35 are not directly involved in channel formation, but they are expected to impair the interface between Sr35 and AvrSr35 and disrupt the assembly of Sr35 resistosome, thereby abolishing its channel activity.

We tested if the CC domain (residues 1 to 150) of Sr35 can trigger channel activity in oocytes, and found that injection of Sr35-CC cRNA did not result in obvious channel currents in TEVC measurement (and no oocyte death as well) (figure 2, a). We then tested whether a C-terminal YFP

tag can be used to mediate proximity induced oligomerization necessary for CC domain cell death function. We found that also here no obvious currents could be measured in TEVC recording (figure 2, b).

Figure 2 Assay of Sr35^{CC} and mutant variants in *Xenopus oocyte*.

- TEVC recordings from *Xenopus oocytes* expressing Sr35^{CC} (residues 1 to 150). The amounts of injected cRNA and the incubation times are indicated. TEVC recordings were performed in ND96 solution, and current amplitudes measured at -110 mV are shown. No oocyte death was observed during incubation in ND96 solution, and no apparent currents were observed in TEVC measurements. Data are mean \pm SEM, $n \geq 8$.
- TEVC recordings from *Xenopus oocytes* expressing Sr35^{CC} and mutant variants, fused with C-terminal YFP. The amounts of injected cRNAs and the incubation times are indicated. TEVC recordings were performed in ND96 solution, and current amplitudes measured at -110 mV are shown. No oocyte death was observed during incubation in ND96 solution, and no apparent currents were observed in TEVC measurements. Data are mean \pm SEM, $n \geq 8$.

Referee #2 (Remarks to the Author):

Forderer et al report the activated form of the plant NLR Sr35 from wheat in complex with the effector AvrSr35 (the resistosome). This is only the second structure of a resistosome of a plant NLR from the CC class (the first one being the resistosome of the Arabidopsis NLR ZAR1). In this way this is a very exciting work that in particular tests the generality of some implications of the ZAR1 structure. Some key new observations brought by this new work involve:

- The first structure of a CC-NLR that recognizes its effector by direct binding (ZAR1 displays indirect recognition)
- The first structure of a CC-NLR from monocots shows that distantly related CC-NLRs from monocots and dicots form pentameric resistosomes
- The similarity extends to the channel structure formed in the resistosome by the CC domain and the corresponding cation channel function.
- Engineering of related barley receptors allowed generation of variants that recognize AvrSr35.
- AvrSr35 structure exhibits a new helical fold.

Having said all this, the paper leaves a feeling that the authors could have made this a more exciting paper, and there are a few disappointing aspects.

- The data on the calcium channel activity and selectivity is not conclusive, and the authors leave in rather unexplained (the text "...the channel activity of the Sr35 resistosome is tolerant to substitutions of acidic residues predicted to line the inner surface of the channel. Thus, we cannot exclude the possibility that the very N-terminus of the Sr35 resistosome (residues 1 to 21) is structurally and functionally distinct from that of the ZAR1 resistosome" more sounds like trying to avoid explaining the data). Why was the L15E/L17E mutant not investigated in the oocyte system?

We added new data for the L15E/L19E mutant in *Xenopus* (Figure 3, c). As expected, this mutant also behaves as a knockout in the *Xenopus* system.

We do not have a clear answer to the question of why the substitutions of the acidic residues in the $\alpha 1$ helix lead to a different outcome in Zar1 and Sr35, nor do we wish to make unfounded speculations in the manuscript. However, it is possible that ion selectivity is mediated by residues other than the tested glutamic acid residues in $\alpha 1$ helix and/or (acidic) residues outside of this region in the inner lining of the extensive Sr35 inner cavity.

- It is not clear if the approach used in section "Steric clash..." is the best approach; why use AlphaFold2 prediction, when one has no control over what conformation of a protein that exists in more than one conformation, such as a plant NLR, is used to derive the predicted model? It would make much more sense to use the inactive ZAR1 structure as a template and derive the model by superimposing the individual Sr35 domains and derive a physically plausible structure by energy minimizing such a model, one would then have much better control over the model in that way. I

suspect the conclusions will be the same, but the approach used here appears somewhat hit-and-miss.

We concur with the point-of-view that structural prediction results from AlphaFold2 is sometimes unpredictable in the outcome. However, we believe that the structure of NLRs can be used to predict whether they are inactive or active. The reason for this is that a comparison of all published structures of inactive NLRs including Arabidopsis ZAR1, tomato NRC1, mouse NLRC4, human NLRP3 and rabbit NOD2 (and the more recently published inactive bovine NLRP9) reveal a highly conserved domain organization of these NLRs. Similarly, the NOD structures of activated NLRs are also highly conserved. The WHD from the NOD module rotates about 180 degrees during NLR activation. These results suggest that the structural organization of the NOD module domains is a defining feature of inactive or active NLRs. We have predicted many NLRs with AlphaFold2 with the setting of always predicting at least five independent models. In most cases, AlphaFold2 predicts NLRs with two conformations in their NOD modules. The two predicted conformations of the NOD modules match well with those seen in the inactive or active NLR structures. The inactive Sr35 structure predicted by AlphaFold2 highly resembles the cryo-EM structure of inactive ZAR1 (Figure 3). As reviewer 3 also correctly pointed out, we made an educated guess in the text. We should have better explained why we believe that the AlphaFold2 prediction is not only trustworthy, but also that it represents the inactive state of Sr35. We made corresponding text changes in the chapter ‘Steric clash between effector and Sr35 NBD mediates receptor activation’ that closely follow the suggestions of reviewer 3.

Figure 3 Structure comparison of Sr35 (prediction) and Zar1 inactive (experimental data). NOD module marked in red for better comparison.

- The abstract mentions a “previously unnoticed arginine cluster” interacting with the EDVID motif; this is incorrect - the structural basis of function of the EDVID motif has been discussed previously in PMID: 31415752, which is not cited in this manuscript

We apologize for not citing this important work by Kobe and colleagues. The reference is now included in 'Sr35 oligomerization and stabilization features are evolutionarily conserved', paragraph 5, line 12. Also, in paragraph 5, line 3, we deleted the part of the sentence 'but its function remained unclear' because the previous study provided correct insight into the function of the EDVID motif. At the same time, we believe our findings that the R-cluster is a conserved feature of CNLs and co-occurs with the EDVID motif remain novelties of our study and we provide experimental data in support of these findings. Therefore, we adjusted the sentence in the abstract as follows:

'Wheat Sr35 and Arabidopsis ZAR1 resistosomes bear striking structural similarity, including an arginine cluster in the LRR domain not previously recognized as conserved, which co-occurs and forms intramolecular interactions with the 'EDVID' motif in the CC domain.'

We hope the reviewer can agree with this text modification.

- The cryo-EM density of the reconstruction is of high quality, but the corresponding structure model appears unfinished and can easily be improved (see details below). When sidechains are not visible, modelling them as alanines or glycines will just lead to confusion by researchers using the model; it would be much better to model the most likely rotamer (or alternatively keep the correct sequence but omit parts of the sidechain that are not visible, although this will undoubtedly also lead to confusion by researchers using the model).

Nevertheless, this is important work and hopefully the authors can improve the paper and the structure models in the final version of the paper.

Problems with the structural model and structural analysis

- Section "Sr35 oligomerization and stabilization features..." paragraph 2: hydrophobic contacts of L47 with Y141 are discussed. This appears to be incorrect, as L47 is not in contact with Y141. The authors may be referring to V46?

- Structure model: AvrSr35

o Residue 130: Appears to be modeled incorrectly.

o Residues 169-170: Missing residues. There is clear map/volume for the main chain in this part and this small chain break could be modeled.

o Residues 171-174: Modeled as polyalanine, not AvrSr35 sequence. This should be modeled as the correct sequence. If uncertain of sidechain positions, either remove sidechains or select most appropriate rotamer.

o Residues 202-206: Modeled as glycine/alanine, not AvrSr35 sequence.

o Residues 206-209: Missing residues can likely be modeled. The map in this region is still clear and distinct at lower threshold.

o Residues 200-209: Poorly modeled.

- o Residues 244-250: Missing residues can likely be modeled. The map in this region is clear and distinct at lower threshold.
- o Residues 273-277: Model slightly out of map register. Fit could be improved.
- o Residues 376-378: Sidechains appear modeled incorrectly.
- o Residues 486-489: Incorrectly modeled.
- Structure model: Sr35 molecule
- o Residues 111-112: Incorrectly modeled. Residues do not fit map, even at low threshold.
- o Residues 346-353: Appears incorrectly modeled. Missing residues 348-352 could be built. There is clear map at lower threshold for the main chain of these missing residues.
- o Residues 564-571: Multiple issues in this region of the structure. Polyalanine is modeled instead of the correct sequence, despite many similar residues (valine and glycine) in the actual sequence. Residues 568-570 are missing, despite the structure appearing as a continuous chain with residue 567 forming a peptide bond with residue 571.
- o Residue 862: Incorrect sequence.
- o Residues 864-868: Sequence is incorrect. Residues modeled as alanine.
- There are many rotamer outliers that are either not supported by the map or have alternative non-outlier positions that are equally well fitting. These outliers should be addressed. Some examples:
 - o AvrSr35: 201, 275, 283, 421, 425, 534.
 - o Sr35: 161, 178, 188, 199, 250, 315, 335, 359, 582, 624, 639, 671, 678, 682, 707, 786.

The problems with structural models have been fixed.

Minor comments on the manuscript

1. Main paragraph 1 line 7: "NODs"

Thank you. We have implemented the change.

2. Section "Sr35 oligomerization and stabilization features..." page 2 paragraph 3 line 4: delete ")"

Thank you. We have made the change.

3. Section "Sr35 oligomerization and stabilization features..." page 2 paragraph 3 line 10: "LRR" stands for "leucine-rich repeat", so "LRR repeats" is redundant.

Thank you for pointing this out. We wanted to exclude potential confusion between the terms 'LRR domain' and 'leucine-rich repeat' (admittedly not in the most elegant way). We changed the sentence to '...each separated by one iteration of the LRR motif, which...'

4. Section “Sr35 oligomerization and stabilization features...” page 2 paragraph 3 line 11 and elsewhere: there is no such thing as “primary (amino acid) sequence” – there is no secondary sequence is there? The correct term is “primary structure”.

We agree and have changed it to ‘Sr35 amino acid sequence’.

5. Section “The Sr35 resistosome has nonselective...” paragraph 1 line 9: “AvrSr35” is misspelled.

Thank you. We have made the change.

6. Section “The Sr35 resistosome has nonselective...” page 2 paragraph 2 line 6 and throughout: “tetrabutylammonium” probably does not have to be capitalized. Check capitalization throughout, “Electrophysiology” section in particular.

We changed the capitalization of ‘tetrabutylammonium’ throughout the manuscript.

7. Materials and Methods section “Cryo-EM sample...” line 11 and elsewhere: put space between number and unit (“50 e/A²”).

Thank you. We have introduced the change.

8. Materials and Methods section “Image processing...” line 4: delete “)”.

Thank you. We have made the change.

9. Materials and Methods section “Image processing...”: why were the two datasets not combined?

We first collected the dataset at Zhengzhou University based on a sample acquired from effector/GST-pull down. The second dataset was acquired from another, completely independent protein preparation (receptor/His-pull down). Initially we hoped to recover a Sr35-AvrSr35 holo-complex versus a pentameric Sr35 apo-complex with the two datasets. However, after analyzing both data sets independently, we realized both purification strategies recovered stoichiometric Sr35-AvrSr35 complexes. We did not combine the two datasets because we had used different purification strategies and because we did not expect a significant improvement of the 3 Å resolution that we had already obtained from the first dataset.

Materials and Methods section “Transient...” page 2 line 1: “sequences”

Thank you. We have introduced the change.

10. Materials and Methods section “Statistical...” line 10: delete underscore.

Thank you. We have implemented the change.

11. References: need to be checked and format made consistent: many are missing the journal (e.g. 6-10 etc), article title and journal title capitalization is inconsistent, journal titles are sometimes abbreviated and sometimes not.

We thank the reviewer for the careful review of the references. We have gone through the references again and made a number of the mentioned changes.

12. Figure 2 legend line 3: “indicates”

Thank you. We have made the change.

13. Figure 2 legend line 16 and elsewhere: “methods” should be “Materials and Methods”.

We changed it here and in legends of Figures 3, 4 and 5.

14. Figure 4 legend line 1: “LRR domain”

Thank you. We have implemented the change.

15. Figure 4 legend line 5-6: “Relative...death” is duplicated.

The duplication has been deleted.

16. Figure 5 legend page 2 line 5: “chimera”

We added the missing ‘a’ to ‘chimer’.

17. Extended Data Fig. 1 legend: there is no “c” (“b” is used twice”) and neither says what is actually shown: chromatogram, gel of some sort?

We corrected the mislabeling of Extended Data Fig. 1 b and c. We also added some more information on what is shown (chromatogram, column, SDS PAGE). In addition, we noticed that in Extended Data Fig.1 we forgot to put an x-axis label and used an outdated label (Sr35ΔOS) instead of the more appropriate Sr35 L15E/L19E label.

18. Extended Data Fig. 5 legend line 5: “Detailed”

Thank you. We have introduced the change.

19. Extended Data Fig. 11 legend line 1: “ROQ1 and RPP1 ligand binding sites”. What is “ascending lateral LRR domain”?

We added labels to ZAR1 explaining the universal definition of LRR surfaces. We agree with reviewer 2 that this definition might not be so familiar with a broader audience and have gladly added the brief graphic explanation.

20. Extended Data Fig. 12: say which structures are predicted and how were they derived.

We added the following sentence: ‘The structure of Sr35 was isolated from the cryo-EM Sr35 resistosome structure, while *HvMLA10*, *HvMLA13*, *TaSH1* and *HvSH1* were predicted using AlphaFold2.’

21. Extended Data Table 1: are these “Selected data from the PDB validation report” , but they are not just validation data, they are structure determination parameters? The lines are misaligned in much of the table.

Indeed, the table legend was somewhat inaccurate. We changed it to ‘Cryo-EM data collection, refinement and validation statistics.’ In addition we made sure that any misalignments in the table itself were removed. The table now also contains the validation statistics of the improved structural model.

Referee #3 (Remarks to the Author):

This manuscript presents a detailed structural analysis of a plant immune receptor-pathogen ligand complex that makes a substantial advance in our understanding of how plants resist disease. The authors find that when bound to its ligand avrSr35, the Sr35 receptor forms a pentameric complex and exhibits cation channel activity. This structure and cation channel activity is similar to that of ZAR1 previously reported by this group, which is the only other structure for an immune receptor of this class (CC-NLR) and derives from a distantly related species. Thus, this work provides evidence that these common features are in fact most likely general characteristics of all or most plant receptors of this class rather than idiosyncrasies of an individual receptor, therefore suggesting common functionalities. Extensive mutational analyses confirm the role of specific residues in effector recognition and oligomerisation as suggested by the structure. Modelling of the inactive state of the receptor further suggests that effector binding to the LRR domain favors the active state through a steric clash with the central NB domain. Swapping of the Sr35 LRR onto distantly related immune receptors allows their activation by AvrSr35, suggesting that this model of receptor activation may be common. Overall, this is a major advance in our understanding of plant immunity, and also adds to the growing understanding of similar classes of immune receptors in humans and other animals and this manuscript can be expected to have substantial and lasting impact in the field.

The methodology applied here is robust, appropriate statistical analyses have been included and the conclusions drawn are based on solid data and reasoning and appear robust. The article is well written and cites prior literature appropriately. I have no suggestions for additional data or experiments, but have made some comments below on some points of interpretation, reasoning, clarification or presentation that the authors should consider.

Some points for the authors to consider.

Lines 118 to 125. In the ZAR1 structures there was a difference between the CC domains in the inactive and active states in the alpha helix 4 region – designated 4a and 4b in the inactive and active structures. CC-alone structures are similar to the inactive ZAR1 state. Could the authors discuss which of these structures is more similar to the Sr35 CC here in terms of the alpha 4 helix? It is hard to tell from the figure, but it looks like there is only a short loop between helices 3 and 4 in this case, which is more like the inactive ZAR1 structure, and those of CC-alone structures. Do the authors think that this change in helix 4 during activation is a common feature of both ZAR1 and Sr35 or perhaps is specific to ZAR1?

We thank the reviewer for investigating the potential differences between the structures of the CC domains of ZAR1 and Sr35. However, we found that helices 2-4 of Sr35 in the resistosome are structurally highly similar to ZAR1 helices 2-4 in the ZAR1 resistosome (Figure 4, a). As for the inactive conformation of ZAR1 and Sr35, we can only rely on the AlphaFold2 prediction of inactive Sr35. Examination of the alignment of the ZAR1 CC experimental structure and the Sr35 CC prediction (Figure 4,b) shows that there is indeed an additional α helix for Sr35 corresponding to a position of α 3 and α 4 helix in Zar1. However, we cannot think of a way to link this predicted additional helix to a function in the Sr35 inactive conformation and thus confirm this part of the Sr35 structure prediction.

Figure 4 Structural alignment of Zar1 (cyan) and Sr35 (green). a CC domain helical barrel from resistosomes. b CC domain of Zar1 inactive structure (PDB 6J5W) and Sr35 inactive prediction generated by AlphaFold2

Line 182-183. Is it necessarily the case that this interaction ‘must’ be disrupted to allow the $\alpha 1$ helix to flip out? Does this assume that the helix itself remains intact during the process? Could not the $\alpha 1$ helix unfold and then refold after moving out? Especially given that this part of the Sr35 structure is not resolved.

This is an interesting point that we have not thought about before. We agree that during the substantial refolding of the CC domain to switch from inactive to active form could require the unfolding of secondary structure elements including α helices. However, unfolding and refolding of the CC domain may occur after it dissociates from the LRR domain. We consider it less likely that the $\alpha 1$ -helix specifically unfolds because it has extensive interactions with the other three $\alpha 1$ -helices of the CC and the NBD. We hope that reviewer 3 can agree that the way we present this hypothesis in the manuscript is merely a suggestion.

Lines 186. This could be rephrased. Rather than the funnel-shaped structure being ‘not well defined’, it is the N-terminal region of Sr35 that is not well-defined while the funnel structure is absent. Eg. “...although the funnel-shaped structure composed of the $\alpha 1$ -helix in the ZAR1 resistosome was not seen in the Sr35 resistosome structure, the corresponding region of Sr35 was not well-defined and apart from this the two structures are highly similar. We thus reasoned...” Could the authors also comment on why this structure is absent – is it likely that this is because of the L15E/L19E double mutation in this region which was used to circumvent cell death in insect cells? Presumably that works by preventing proper function of the $\alpha 1$ helix, without affecting structure of the rest of the protein protein or its ability to recognise the effector and transform to an active state as observed here. It seems likely that those mutations could primarily affect funnel formation.

We rephrased the sentence in line 186 to hopefully accommodate the criticism.

We agree that the L15E/L19E substitution could be responsible for the lack of a cryo EM density in this region. At the same time, α helices are formed by hydrogen bonding involving the amino acid backbone rather than specific side chains. Following this logic, it would mean that L to E substitutions can act as a ‘helix breaker’ similar to amino acids like proline. Another idea as to why we are missing this part of the structure could be the N-terminal GST tag. However, our Zhengzhou dataset, which was used to solve the cryo-EM structure of the Sr35 resistosome, was collected from the sample with the N-terminal GST being removed by protease cleavage. Nevertheless, there are still several residues covalently connected to the N-terminus of Sr35 after GST is removed. It is possible that these residues make the N-terminal $\alpha 1$ -helix flexible in the structure. Alternatively, Sr35 may require a membrane context for rigid folding of this region, a context that we did not provide by purifying the protein in aqueous solution. We can only speculate why the ZAR1 amino-terminal region is more rigid in solution.

Lines 202-204. Some further explanation of the reasoning here would be useful. Is the A01 inhibitor expected to completely block all endogenous xenopus Ca channel activity? Hence the remaining activity is due to the Sr35 channel not being affected by this inhibitor due to its non-canonical nature? LaCl₃ is a non-specific inhibitor as a Ca mimic?

We thank reviewer 3 for the suggestion and questions. We hope it is acceptable that we interpreted the question as follows: “Is the A01 inhibitor expected to completely block all endogenous oocyte Ca²⁺-activated chloride channel activity”.

The CaCCinh-A01 inhibitor is a potent calcium-activated chloride channel (CaCC) blocker. In a study in CHO cells, it has an IC₅₀ of ~ 7.84 μM and ~ 90.67% inhibition at 300 μM concentration (Liu et al. 2014). In the current experiments, we expect CaCCinh-A01 to significantly block endogenous oocyte calcium-activated chloride channel, similar to the levels observed in CHO cells (not complete inhibition). The residual current after treatment with CaCCinh-A01 inhibitor is largely contributed by the oocyte-expressed Sr35 channels, which can non-selectively conduct cations such as Na⁺, K⁺, Mg²⁺ and Ca²⁺.

La³⁺ is a mimic of Ca²⁺, and LaCl₃ is usually used as a non-specific Ca²⁺ channel blocker.

*Liu, Y., Zhang, H., Huang, D., Qi, J., Xu, J., Gao, H., et al. (2014). Characterization of the effects of Cl⁻ channel modulators on TMEM16A and bestrophin-1 Ca²⁺ activated Cl⁻ channels. *Pflügers Archiv - European Journal of Physiology*, 467(7), 1417–1430. doi:10.1007/s00424-014-1572-5

Line 256-260. This section seems to be presented in a manner that proceeds from an assumption, rather than analysing a result and making a conclusion. What is the justification for claiming this is a model of inactive Sr35? What confidence do we have that it is correct? Is it similar to the inactive state of ZAR1? I think this should be presented in a different logical order along the lines of the following – the authors could express this more precisely I'm sure: "We used alphafold2 to predict a structure for Sr35 in the absence of AvrSr35 and compared this to the resistosome structure. Many parts were similar (eg LRR), suggesting that alphafold did a good job of local structure prediction, but the NOD module showed substantial organisational differences. ((The modelled structure was similar to the ZAR1 inactive state (???), suggesting that it is likely to be a good representation of the Sr35 inactive state – note that if it is not similar to inactive ZAR1, then we'd be concerned about whether it is correct.)) Modelling of AvrSr35 onto the LRR domain of the predicted Sr35 structure shows substantial overlap between the effector and the Sr35 NBD. Assuming the modelled structure represents the inactive state of Sr35, this steric clash..."

We thank reviewer 3 for this very valid point. We agree that we jumped to the conclusion in the original text version rather than starting from an unbiased way of thinking. We have rewritten the paragraph following the suggestion closely and thank you for drafting an improved version of this text section.

Line 267-268. Replace 'appears a conserved mechanism of plant NLRs' with " is a common feature of the four known resistosome structures (ED fig 11), suggesting it may be a conserved mechanism of plant NLRs." Given this conclusion, could the authors elaborate a little on why this should be so? It seems to me that the 'lateral ascending side' of the LRR is the one that is adjacent to the NOD region in the known structures. Thus, effector binding here is more likely to lead to a steric clash with the inactive conformation. Is this a reasonable argument to propose? Binding on the other side could potentially occur irrespective of the activation state, hence would not predispose the receptor towards either state.

We agree with this insight and have clarified the sentence.

We also added a new reference of a review that we have published in the meantime. In it, we go in more detail about the activation mechanism and ligand binding to the ascending lateral side of plant NLRs.

In our opinion, this region is evolutionarily favored for ligand binding because it is exposed to the surrounding solvent while ensuring a steric clash with the NBD of the ligand. By contrast, the concave side of the LRR (which is the more typical LRR binding surface in e.g. VLR antibodies of lampreys), is obstructed for ligand entry by the NBD domain. Of course, this only holds true assuming the LRR ligand is a rather sizeable protein, but smaller ligands might not obey to this 'rule'. However, regardless of the ligand sizes, binding to the other side would have no impact on the activation state of NLRs, as the reviewer commented. We wish to mention that the mechanism most likely applies to singleton NLRs. Whether paired NLRs, consisting of sensor and helper NLRs, follow this 'rule' remains to be investigated. We would not be surprised if evolutionary processes have generated alternatives based on this NLR activation blueprint.

Line 317 to 320. The reasoning presented here could be refined a little. Firstly, it should be suggested that 'the activation and signalling mechanisms of CNLs are evolutionarily conserved' (we know the CNLs themselves are evolutionarily related). Of the 'Two lines of evidence' discussed, one refers to exchanges between wheat and barley NLRs. Since these are both monocots it does not bear on the conservation between monocots and dicots. Conversely, the subsequent statement that 'this conservation extends to cation fluxes' constitutes an additional (third) line of evidence. Perhaps just delete 'monocotylenous and dicotyledonous' and then cite three lines of evidence. The key part of the wheat Sr35 and barley MLA swap is not so much that it is between species, but that the genes are not orthologous. Swapping between wheat and barley orthologs may not be much different than between alleles, which has been done extensively previously, but this represents a further step as the NLRs are less related.

We rephrased the section as follows:

'Our results, together with earlier data, strongly suggest that the activation and signalling mechanisms of CNLs are evolutionarily conserved. Three independent lines of evidence support this idea: our structural elucidation of the wheat Sr35 resistosome and its similarity to the previously reported Arabidopsis ZAR1 resistosome structure². Second, the functional interspecies hybrid receptors generated from the non-orthologous CNLs wheat Sr35 and barley MLAs. Third, the conservation extends to the non-selective cation flux across membranes enabled by pentamerization, although it is possible that ion selectivity and channel dynamics differ between individual CNLs,...

Minor edits.

Line 55. Reference 15 Yu et al 2016 describes genetic mapping of Sr46 not the gene cloning. Should cite here Arora et al. 2019, Resistance gene cloning from a wild crop relative by sequence capture and association genetics. Nature biotechnology, 37, 139–143. <https://doi.org/10.1038/s41587-018-0007-9>

Thank you. We have made the change.

Line 66 and line 73. Reference to Fig1a should be to extended data fig 1a.

We apologize for the mistake. It should be Extended Fig 1a.

Line 69. The Sr35-L15E/L19E mutant is not named in ED fig 1a. Is this the sample labelled deltaOS in the figure?

Please excuse the error. In our laboratory record, the L15E/L19E mutant was referred to as 'deltaOS' for a long time. We made the correction in the figure.

Line 78. This should be rephrased as the UV absorbance is not 550kDa (70 mL), but the peak of UV absorbance occurs at an elution volume of 70mL corresponding to a size of 550kDa. However, please check this versus the figure which seems to show the peak at about 65 mL and labels this with an arrow indicating a size of 900kDa.

Thank you for spotting this discrepancy. We have changed the text to match the figure:

'In SEC, the affinity-purified protein complex eluted with a broad peak with a maximum UV absorbance at 65 mL elution volume exceeding the 669 kDa (66 mL) of our largest protein marker.'

Extended data figure 1c. Not all of the lanes are labelled. Mostly it would seem that the missing ones are continuous ie 63 between 62 and 64, but this is not so for all the lanes at the right.

This is because the numbers are all consecutive, so we have written down only every second number to avoid overcrowding. This is also true for the lanes on the right, except for the last lane, which is the 84 mL fraction at the peak of the AvrSr35 monomer elution. To avoid any confusion from the labelling we added the 74 mL label to the second last label.

Line 80 reference to Figure 1a should also include fig 1b

Line 84 reference to figure 1b here does not seem relevant.

To the above two comments: Figure 1b are 2D averages from cryo-EM data and referenced in line 84, while Figure 1a is a raw image from negative staining EM. Cryo-EM and negative staining are two different ways of sample preparation. Negative staining was used as a quality check for a protein preparation and we used uranyl acetate solution, which causes damage to the native protein structure and cannot be used for structure determination at high resolution. On the other hand, cryo-EM sample preparation aims at preserving the native folding of a protein. However, due to the lack of a contrast giving agent (e.g. uranyl acetate), contrast in raw images tends to be very low and 2D averages are more commonly shown. In light of this explanation, we hope that reviewer 3 can agree that our original figure referenced here is correct.

Line 139 and Fig 2g. The text states that an L42E mutation was made, but the protoplast section of the figure is labelled L42A, L42E is in the benthamiana part of the figure. ED Fig4a also has L42A

We made the label consistent between text and figure. It should have been L42E throughout.

Line 164. Sr35 should be ZAR1.

Thank you for spotting the writing mistake. We changed it to ZAR1.

Line 150-170 – note that the Rairdan et al paper that defined the EDVID motif (ref 24) also actually showed that it had a role in mediating interaction between the CC domain and the NB-LRR, which is consistent with the structural findings here.

As reviewer 2 also correctly pointed out, we had forgotten to acknowledge earlier work on the EDVID motif. We made some changes to the paragraph lines 150-170 and also edited the corresponding sentence in the abstract:

‘Wheat Sr35 and Arabidopsis ZAR1 resistosomes bear striking structural similarity, including an arginine cluster in the LRR domain not previously recognized as conserved, which co-occurs and forms intramolecular interactions with the ‘EDVID’ motif in the CC domain.’

Line 174. Fig 2h,i should refer to fig 2i,j.

Thank you. We have introduced the change.

Line 177 delete respectively.

Thank you. We have implemented the change.

Line 221. The term ‘ascending lateral side’ may not be clear to non-LRR specialists.

For explanation, we have included a small graphical representation in ED Fig 11.

Line 251. This is a little vague, but it would be worth to point out more specifically that most of the single amino acid mutations had no effect on AvrSr35 recognition, while the single mutations Y387A and R395A did have a partial effect on reducing recognition but did not abrogate it entirely. Also the Q72* mutant (a premature stop codon?) is not mentioned.

We have rewritten the text here to make the partial effect of Y387A and R395A mutants clearer. We also added an explanation about the premature stop codon mutant Q72* in line 249 and made a reference to Salcedo et al 2017 who first identified this allele of AvrSr35.

We also noticed a labeling mistake: A383Y/A388Y should have been A384Y/A388Y throughout the manuscript. We made the necessary changes.

Line 283, 296. These are not ‘single’ substitutions. There were either 8 or 10 substitutions made in each case. Perhaps rephrase as “targetted mutations of specific residues”.

Thank you for spotting this inaccuracy. We changed ‘single amino acid substitutions’ to ‘targeted amino acid substitutions’

Fig2 J “Y74A/E77A/D78A” – the A is missing after Y74 in this label.

Thank you for spotting this error. We made the change.

Figure 4 legend, repeated sentence “Relative luminescence was used as a readout for cell death”

The duplicate was removed.

ED Fig1b,c figure legend is not informative eg fig1b shows a size exclusion chromatography profile. And 1c (labelled b again in legend) shows a stained electrophoresis gel.

We changed the labelling in ED Fig1 and also added some more detail in the figure legend.

Referee #4 (Remarks to the Author):

In this study, Förderer, Li, Lawson, Deng and colleagues developed a high-resolution structure of the wheat NLR immune receptor protein Sr35 in complex with its corresponding effector AvrSr35. They find that the active immune receptor complex forms a higher order pentameric complex which has structural features similar to that of the recently resolved ZAR1 complex from the model plant *Arabidopsis thaliana*. The authors performed frog oocyte electrophysiological assays to show that the complex acts as a non-selective cation channel. In a series of elegant experiments, the authors then used the structure to guide the engineering of Sr35 homologues from wheat and barley which do not recognize AvrSr35. They demonstrate that a small number of amino acid substitutions in Sr35 (which are surface exposed and make contact with AvrSr35) are enough to convert these wheat and barley homologues into gain-of-function variants that mount an effective AvrSr35-mediated immune response.

This is a tour-de-force study encompassing cutting edge cryo-electron microscopy. We learn that coiled-coil NLRs separated by 200 million years of evolution (i.e. defining the major split in land plants between monocots and dicots) form similar oligomeric immune complexes. The deft structure-guided approach to engineering new-to-nature functional immune receptors has potential important applications in wheat disease resistance breeding as the few amino acid changes required (8 to 10) would allow these variants to be introduced into the wheat genome by gene editing and subsequently deployed in several jurisdictions including the USA, Canada, Australia, China and South America, where these type of biotech crops are now accepted.

The manuscript is exceptionally well written with highly didactic figures. I expect the study will gain the strong attention of the communities studying molecular plant-microbe interactions, cereal disease resistance, wheat breeding, crop protection biology, genetic engineering, and structural biology. In conclusion, I am highly supportive of this study. In making suggestions for improvements, I have only a few minor comments, which can all be addressed without the need for additional experimentation, as noted below:

1. The study has generated a large number of constructs. Please consider making these available in Addgene as this will be a great resource for the scientific community.

We are currently looking into sending our constructs to Addgene. While we are very motivated to do so, please keep in mind that it will take some time to prepare all the stocks and we do not wish to withhold the manuscript until we have completed this task.

2. Page 2: The authors state that: “Wheat stem rust caused by the fungal infection with *Puccinia graminis* f. sp. *tritici* (Pgt) threatens global food production [...] has motivated the search for stem rust resistance genes [...]. This resulted in the isolation of eight stem rust resistance (Sr) genes that are phylogenetically related and belong to a clade of grass CNLs all of which confer strain-specific immunity⁹⁻¹⁵.”

In response to this statement please note the following:

(i) The citation should also include reference number 16 which describes the cloning of Sr26 and Sr61 which encode CNLs. If I am not mistaken, that brings the total tally of to ten i.e. Sr13, Sr21, Sr22, Sr26, Sr33, Sr35, Sr45, Sr46, Sr50 and Sr61.

We added the reference Zhang et al. 2021 to this list.

(ii) You might also consider including SrTA1662, cloned from *Aegilops tauschii*, and introgressed into wheat, see Gaurav et al., 2022: <https://www.nature.com/articles/s41587-021-01058-4>

This would bring the total tally to 11.

We added the reference. Many thanks for providing us with the state-of-the-art information on Sr breeding.

(iii) The reference for Sr46 cloning (Yu et al., 2015) is wrong. This paper provides genetic characterization of Sr46. The cloning paper is Arora et al., 2019: <https://pubmed.ncbi.nlm.nih.gov/30718880/>

Thank you for spotting the error. We changed the reference.

(iv) The way the sentence is currently phrased one is led to believe that all the cloned Sr genes are CNLs. However, the same motivation that led to the cloning of the CNL Sr genes also led to the cloning of Sr60 and Sr62, which in fact encode wheat-tandem kinases, see:

<https://pubmed.ncbi.nlm.nih.gov/31487050/>

<https://pubmed.ncbi.nlm.nih.gov/35338132/>

We prefer to focus on the CNLs in this list because Sr35, after all, is a CNL. However, we have changed the sentence slightly to avoid giving the impression that all Sr genes are CNLs. ‘This resulted in the isolation of eleven phylogenetically related Sr genes that belong to a clade of grass CNLs all of which confer strain-specific immunity.’

(v) Finally, please note that “f sp.” should be “f. sp.”

We thank reviewer 4 for this information and have changed ‘f sp.’ to ‘f. sp.’.

3. On page 3, the authors introduce the experiment in which Sr35 and AvrSr35 are expressed in insect cells. They go onto state: “Unexpectedly, cell death was observed when the receptor was co-expressed with AvrSr35 (Fig. 1a)...” A reference to this result is also made in the Discussion on page 10. However, I could not see these insect cell death results depicted in Figure 1a. Perhaps the authors meant to cite Supplementary Figure 1a? In this figure, please define explicitly the y-axis.

This is correct. We wanted to refer to Extended Data Fig 1a. The mistake was corrected and we also corrected labeling mistakes including the missing y-axis label.

4. On page 4, “known proteins structures” should be “known protein structures”.

This has been corrected.

5. Top of page 7. Please consider spelling out the abbreviation/acronym of “CaCC” – I think this is the first use of the abbreviation.

We introduce the abbreviation ‘calcium-gated chloride channels (CaCC)’ in line 200. We then use CaCC inhibitor A01 in line 202. We believe it should be correct as it is.

6. Page 8. “with Sr35 LRR” should be “with the Sr35 LRR”.

Thank you. We made the change.

7. Figure 3, page 24. The figure legend starts with the abbreviation TEVC – please consider spelling this out in full.

We spelled it out here.

8. Figure 4. Why is the R395A mutation (tested in *N. benthamiana* in panel f) not highlighted in panel b?

Thank you for spotting this. We agree that showing residue R395 in Fig 4b should be informative and have added the residue in the revision.

Reviewer Reports on the First Revision:

Referees' comments:

Referee #1 (Remarks to the Author):

The revision is satisfactory. Congratulations on the excellent work!

Referee #2 (Remarks to the Author):

I believe the authors have adequately addressed the comments and the manuscript is ready for publication.

Referee #3 (Remarks to the Author):

I think the authors have done an excellent job in responding to the review comments and the revised manuscript is improved and suitable for publication. This is a really important contribution to understanding NLR immune receptor function.

Referee #4 (Remarks to the Author):

The authors have addressed all the concerns I raised with the initial submission. I have no further concerns. BW

Post-review note from authors:

In Fig. 1, on page 13, in the top SDS page, L11E/L15E should have read L15E/L19E.

On page 13, lane 3, it should have read 'to bind Sr35 receptor/receptor variants indirectly to glutathione beads...'